# Single-cell RNA-sequencing of peripheral blood mononuclear cells reveals widespread, context-specific gene expression regulation upon pathogenic exposure

Roy Oelen [1,2,11], Dylan H. de Vries [1,2,11], Harm Brugge[1,2,11], M. Grace Gordon [3,4,5,6], Martijn Vochteloo [1,2], single-cell eQTLGen consortium*, BIOS Consortium*, Chun J. Ye [4,6,7,8,9,10], Harm-Jan Westra[1,2], Lude Franke[1,2,12 ✉] & Monique G. P. van der Wijst [1,2,12 ✉]

The host's gene expression and gene regulatory response to pathogen exposure can be influenced by a combination of the host's genetic background, the type of and exposure time to pathogens. Here we provide a detailed dissection of this using single-cell RNA-sequencing of 1.3M peripheral blood mononuclear cells from 120 individuals, longitudinally exposed to three different pathogens. These analyses indicate that cell-type-specificity is a more prominent factor than pathogen-specificity regarding contexts that affect how genetics influences gene expression (i.e., eQTL) and co-expression (i.e., co-expression QTL). In monocytes, the strongest responder to pathogen stimulations, 71.4% of the genetic variants whose effect on gene expression is influenced by pathogen exposure (i.e., response QTL) also affect the co-expression between genes. This indicates widespread, context-specific changes in gene expression level and its regulation that are driven by genetics. Pathway analysis on the *CLEC12A* gene that exemplifies cell-type-, exposure-time- and genetic-background-dependent co-expression interactions, shows enrichment of the interferon (IFN) pathway specifically at 3-h post-exposure in monocytes. Similar genetic background-dependent association between IFN activity and *CLEC12A* co-expression patterns is confirmed in systemic lupus erythematosus by in silico analysis, which implies that *CLEC12A* might be an IFN-regulated gene. Altogether, this study highlights the importance of context for gaining a better understanding of the mechanisms of gene regulation in health and disease.

[1] Department of Genetics, University of Groningen, University Medical Center Groningen, Groningen, The Netherlands. [2] Oncode Institute, Utrecht, The Netherlands. [3] Biological and Medical Informatics Graduate Program, University of California San Francisco, San Francisco, CA, USA. [4] Institute for Human Genetics, University of California San Francisco, San Francisco, CA, USA. [5] Department of Bioengineering and Therapeutic Sciences, University of California San Francisco, San Francisco, CA, USA. [6] UCSF Division of Rheumatology, Department of Medicine, University of California San Francisco, San Francisco, CA, USA. [7] Bakar Computational Health Sciences Institute, University of California San Francisco, San Francisco, CA, USA. [8] Department of Epidemiology and Biostatistics, University of California San Francisco, San Francisco, CA, USA. [9] Parker Institute for Cancer Immunotherapy, San Francisco, CA, USA. [10] Chan Zuckerberg Biohub, San Francisco, CA, USA. [11] These authors contributed equally: Roy Oelen, Dylan H. de Vries, Harm Brugge. [12] These authors jointly supervised this work: Lude Franke, Monique G. P. van der Wijst. *Lists of authors and their affiliations appear at the end of the paper. ✉email: l.h.franke@umcg.nl; m.g.p.van.der.wijst@umcg.nl

Over a decade of genome-wide association studies (GWAS) has revealed thousands of genetic variants associated with disease risk[1], most of them single nucleotide polymorphisms (SNPs). Despite this, the cascade of events through which these variants change disease risk remains largely unclear. One way to dissect this cascade is by linking disease-associated SNPs to downstream gene expression through so-called expression quantitative trait locus (eQTL) analysis[2]. However, recent work by Yao et al. indicated that, on average, only $11 \pm 2\%$ of disease heritability is mediated by cis-eQTLs, i.e., SNPs affecting the expression of nearby genes[3]. One explanation for this relatively low contribution could be that many of these eQTL effects are cell-type-specific and context-dependent[4,5], which means that their disease contribution cannot be accurately estimated using the steady-state expression in bulk-averaged tissues. In other words, the relevant context for a particular disease-associated SNP may not have been studied yet, meaning that many of the true downstream effects of these SNPs remain hidden[6]. In a first effort to identify tissue-specific eQTLs, the GTEx consortium performed eQTL analysis in 44 different human tissues across 449 individuals (70–361 individuals/tissue)[7]. However, this study was limited by the relatively small number of donors for many of the tissues and the lack of cell-type-specific resolution. More recently, with the advent of high-throughput, cost-efficient single-cell RNA-sequencing (scRNA-seq) technologies[8,9], it has become possible to assess both the cell-type-specific and context-dependent effects of risk SNPs on downstream gene expression[10–12].

While the tissue or cell type is one context that can affect the association between a SNP genotype and gene expression, many other contexts can also be of influence. For the immune system, for example, exposure to specific pathogens commonly occurs and the immune response following exposure can create the environmental context required to change specific interactions between genetics and downstream gene expression[4,13–17]. In turn, these context-specific interactions may explain why exposure to specific pathogens has been associated with the development of autoimmune diseases in individuals with a genetic predisposition[18]. For example, reovirus can disrupt intestinal immune homeostasis and initiate a loss of tolerance to gluten in individuals expressing HLA-DQ2 or HLA-DQ8, leading to celiac disease[19]. Another example is the strong indications that enteroviral infections in the pancreas, such as with coxsackievirus, in genetically predisposed individuals may accelerate the development of type I diabetes (T1D)[20–22]. Several T1D-associated risk genes affect the antiviral response through the regulation of type I interferon (IFN) signaling[23]. When the insulin-producing pancreatic β cells of genetically predisposed individuals are then exposed to such viruses, incomplete viral clearance and chronic infection of these β cells may be the consequence. This could then induce β cell apoptosis that contributes to the development of T1D[24,25]. Overall, it is estimated that 11–30% of autoimmune risk loci involve cis-eQTLs in blood, and it is hypothesized that trait-associated eQTLs have increased context-specificity[26–28]. Given this hypothesized context-specificity, it is important to study eQTLs in a variety of different contexts to determine the possible effect of the environment on the interplay between genetic variation and gene expression in disease.

This study aims to disentangle the gene expression and gene regulatory processes that are driven by differences in genetics and/or pathogen exposures, and that could explain how inter-individual differences can contribute to disease risk. Moreover, we show how the properties of scRNA-seq data (i.e., cell- and context-specific resolution, a high number of cellular observations per individual) can be employed to disentangle the molecular mechanisms that underlie the context-specificity of the genetic regulation. By disentangling these mechanisms, we provide novel insights into how genetics can contribute to disease risk aiding us to reduce such risk in the future.

## Results

**Single-cell profiling of immune cells upon pathogen stimulation.** Here we present the 1M-scBloodNL study in which we performed 10x Genomics scRNA-seq on 120 individuals from the Northern Netherlands population cohort Lifelines. For each individual, we sequenced peripheral blood mononuclear cells (PBMC) in an unstimulated condition and after 3 h and 24 h in vitro stimulation with C. albicans (CA), M. tuberculosis (MTB), or P. aeruginosa (PA), totaling ~1.3 million cells (Fig. 1, Supplementary Data 1). A combination of 10x Genomics v2 and v3 chemistry reagents was used to capture an average of 1226 cells per individual per condition (v2: 907 genes/cell, v3: 1861 genes/cell) (Supplementary Data 2). Souporcell[29] was used to identify the doublets coming from different individuals, followed by sample demultiplexing using Demuxlet[11]. This revealed, on average, 12.0% of cells as doublets. Due to differences in gene amplification between v2 and v3 chemistry, determination of quality control (QC) thresholds and analyses were performed separately per chemistry (Supplementary Fig. 1a–d). Results from both chemistries were then meta-analyzed for interpretation. Low-quality cells were excluded, leaving 928,275 cells in the final dataset used for analysis (see Methods, Supplementary Data 3). UMAP dimensionality reduction and KNN-clustering were then applied to the normalized, integrated count data, allowing the identification of six main cell types: B, CD4+ T, CD8+ T, monocytes, natural killer (NK), and dendritic cells (DCs) (Supplementary Fig. 1e–g, Supplementary Data 4), for which the latter five were further subdivided in two subcell types each: naïve and memory CD4+ T and CD8+ T cells, classical (cMono) and non-classical monocytes (ncMono), NKdim and NKbright, myeloid (mDC), and plasmacytoid DCs (pDC).

**Gene expression response upon pathogen stimulation reveals stronger cell-type-specificity than pathogen-specificity.** To assess the transcriptional changes upon pathogen stimulation with CA, MTB, and PA, we performed differential expression (DE) analysis using MAST in each of the major cell types and their subcell types (Supplementary Data 5)[30]. For the major cell types, pairwise comparisons between the untreated and pathogen-stimulated conditions revealed between 688 and 2022 DE genes after 3 h stimulation, further increasing from 1052 to 2616 DE genes after 24 h stimulation (Fig. 2a). The number of DE genes was comparable between the different pathogen stimulations at the same timepoint but differed strongly between some cell types. Myeloid cells (monocytes and DCs) showed the highest number of DE genes, whereas both CD4+ and CD8+ T cells showed the fewest DE genes. This is consistent with the innate immune cells being the first responders during pathogen stimulation[31].

A total of 5516 unique DE genes were identified across all conditions and major cell types, and an additional 1621 DE genes were identified in the subcell types (Supplementary Data 5). This indicates that most DE genes can already be identified at the major cell-type level. However, since the statistical power to detect such DE effects is correlated with the number of cells within a subcell type[32], likely some of the subcell type specificity remains undetectable. Of the 5516 DE genes within the major cell types, 31.1% were cell-type-specific and 15.1% were shared across all major cell types (Fig. 2b). The fraction of DE genes that were cell-type-specific was comparable for each of the cell types, but, in absolute numbers, monocytes and DCs had the most unique DE genes. Sharing between different pathogen stimulations at the

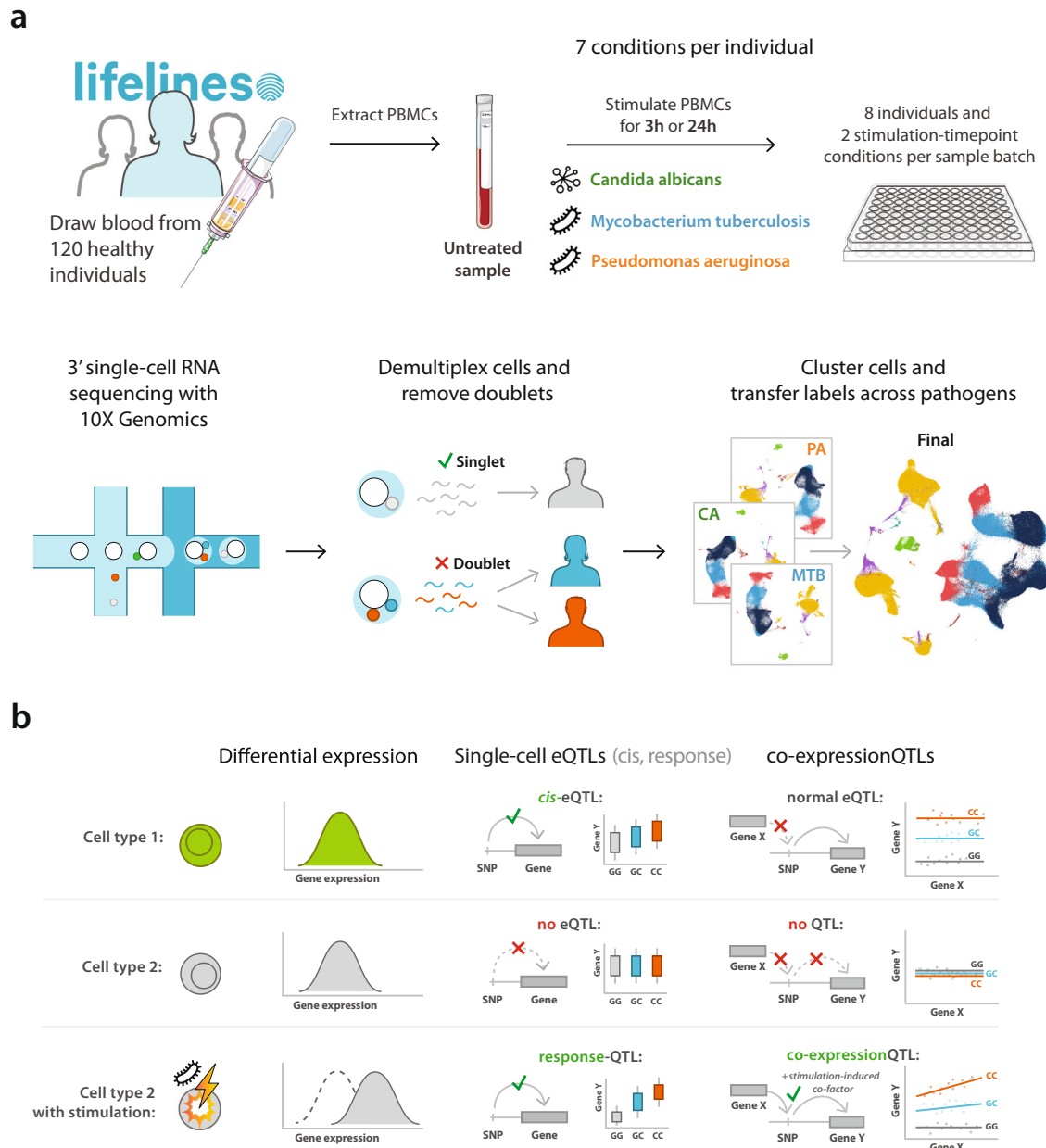

**Fig. 1 Study overview. a** Blood was drawn from 120 individuals from the Northern Netherlands participating in the population-based cohort Lifelines. PBMCs were isolated within 2 h of blood collection and cryopreserved in liquid nitrogen until further use. For each scRNA-seq experiment, PBMCs from 16 individuals were thawed. Per individual, these PBMCs were left untreated (UT) or were stimulated with *C. aAlbicans* (CA), *M. tTuberculosis* (MTB), or *P. aAeruginosa* (PA) for 3 h or 24 h in a 96-well plate. In total, this resulted in 7 stimulation–timepoint combinations per 120 individuals = 840 different conditions processed. Multiplexed 3′-end scRNA-seq was performed using the v2 and v3 chemistries of the 10x Genomics platform. Per experiment, two sample batches of a 10x chip were loaded, each containing a mixture of eight individuals and a combination of two different stimulation–timepoint combinations. After sequencing, samples were demultiplexed and doublets were identified. Cell-type classification was performed on the QCed dataset by clustering the cells per pathogen, mixed with cells of the unstimulated condition. The cell-type labels were subsequently transferred back to the dataset containing all cells. **b** This study was conducted to identify cell-type-specific or pathogen-stimulation-dependent: (1) differentially expressed genes, (2) eQTLs and response-QTLs, and (3) co-expression QTLs.

same timepoint was more prominent than sharing between different timepoints within the same pathogen stimulation (Fig. 2c, Supplementary Fig. 1f): 39.8% of the total unique DE genes were shared across the same timepoint (7.4% at 3 h and 32.4% at 24 h), whereas only 10.3% of DE genes were unique to a specific pathogen stimulation and 41.3% were shared across all stimulation–timepoint combinations. This indicates that the immune response to our pathogen stimulations of both bacterial and fungal origin was more specific to timepoint after stimulation

than to the type of pathogen. Consequently, the genetic control of these responsive genes is expected to be more time-dependent than pathogen-dependent.

To evaluate the DE results and confirm proper activation of the cells upon stimulation, we performed two different analyses. In the first analysis, we measured the activity of a general stimulation-responsive pathway—the antigen-processing cross-presentation pathway (REACTOME R-HSA-1236975)—that should become activated in each of the cell types and upon each

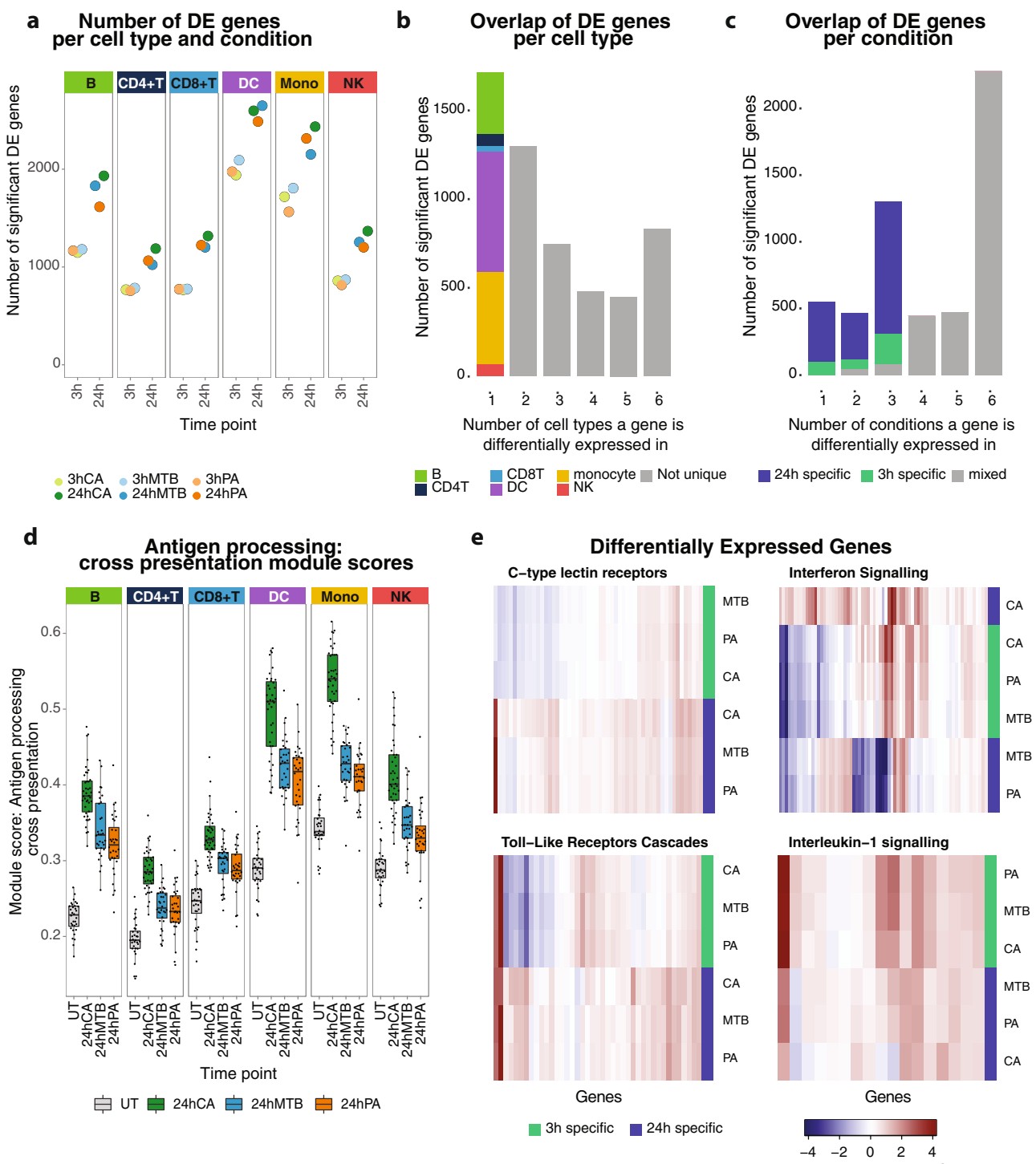

**Fig. 2 Differentially expressed genes and pathways upon pathogen stimulation. a** Number of DE genes per cell type upon 3 h (light colors) or 24 h (dark colors) stimulation with *C. aAlbicans* (CA, green), *M. tTuberculosis* (MTB, blue), or *P. aAeruginosa* (PA, orange). **b** Bar plot showing the overlap of DE genes across cell types. The first bar, depicting the cell-type-specific DE genes, is colored based on the cell type in which the DE gene is found. **c** Bar plot showing the overlap of DE genes across pathogen–timepoint combinations (3 h vs 24 h stimulation with CA, PA or MTB). Bars are colored based on the length of stimulation. **d** Boxplots (showing median, 25th and 75th percentile, and 1.5× the interquartile range) representing the module score of the antigen-processing cross-presentation pathway across all individuals per cell type and per 24 h pathogen-stimulated condition. Each dot shows the average module score per individual (V3 chemistry is shown, the Source Data file includes the individual data points). **e** Heatmaps showing the immune-related DE genes in monocytes (V3 chemistry is shown), split by their involvement in either one of the four selected immune pathways associated with pathogen recognition and its downstream signaling. C-type lectin and toll-like receptors show more general activation upon pathogen stimulation, whereas interleukin-1 and interferon signaling show a more specific expression pattern with timepoint (3 h stimulation) or stimulation (CA), respectively. DE summary statistics can be found in Supplementary Data 5. The number of individuals and cells included in each analysis can be found in the Source Data file.

of the pathogen stimulations. This analysis revealed increased activity of the antigen-processing pathway-associated genes across all cell types after 24 h stimulation and for each of the pathogens (Fig. 2d). In the second analysis, we focused on DE genes identified upon 24 h stimulation with CA. We had previously performed similar analyses in a smaller scRNA-seq study[15], so we could use this study for comparison purposes. This analysis revealed a high concordance between DE genes in our current study and those from our previous study, varying from 73% for the monocytes up to 93% for the B cells (Supplementary Fig. 2). In general, these analyses showed that CA stimulation resulted in the highest activation of genes associated with the antigen-processing pathway and that monocytes were the cell type with the strongest response. These two analyses confirmed proper activation of the cells and stimulation responses that were in line with previous literature[15,33].

Next, we determined which pathways were enriched within the upregulated DE genes for each cell type and each pathogen–timepoint combination (Supplementary Data 6). In line with the DE results, most of the enriched pathways were shared across the different pathogen-stimulation conditions within the same time-point (Fig. 2e). To highlight relevant pathways involved in pathogen recognition and downstream immune response, we filtered the enriched pathways for those related to the 'Immune system' REACTOME pathway parent term. For this illustrative example, we selected monocytes because this was the cell type in which we observed the most DE genes (Fig. 2e). Here we observed a general activation of pathogen-recognition receptors and downstream signaling, including the C-type lectin and toll-like receptors. Some pathways, such as interleukin-1 (IL-1) signaling, were clearly enriched at a specific timepoint (3 h stimulation), whereas others, such as the IFN pathway, showed a notable difference between different pathogen stimulations (more prominently activated in CA compared to the other two pathogens). These findings corroborate literature describing IFN as an important signaling pathway in response to all three pathogens[33–35] and that IL-1 family molecules are part of the early stages (<14 h) of the inflammatory response in monocytes with their expression decreasing again in later stages[36].

For the subcell types, we were mainly interested in those pathways that were differentially activated upon pathogen stimulation between the two subcell types of each major cell type. For this, we visualized the top 10 most enriched pathways with the largest difference in significance between both subcell types (Supplementary Fig. 3). This revealed that most pathways were enriched in both subtypes, but that the relative activation could differ. For example, several pathways associated with interferon signaling were more significantly enriched in the ncMono as opposed to the cMono (Supplementary Fig. 3, Supplementary Data 6).

**The number of eQTLs decreases in cells with a stronger stimulation response**. Our experimental setup, in which we analyzed pathogen-stimulated PBMCs using scRNA-seq, allowed us to investigate the extent to which SNPs affect gene expression in different contexts. To maximize the power to detect eQTLs, we took advantage of a previously conducted genome-wide cis-eQTL meta-analysis in 31,684 whole-blood bulk samples (eQTLGen[37]) by only testing their top SNP–gene combinations, i.e., lead-eSNPs. Due to the power of eQTLGen, they could identify even cis-eQTL effects with a small effect size. We therefore expected that many of the context-specific effects, to which only a subset of individuals or cell types might have been exposed, should have resulted in an eQTL effect identified in eQTLGen. However, compared to the eQTLGen bulk whole-blood dataset, our pathogen-stimulated

scRNA-seq data have the additional benefit that it can identify the cell types and contexts in which these eQTL effects manifest themselves.

We performed the eQTLGen lead-eSNP cis-eQTL discovery analysis per cell type and for each stimulation–timepoint combination separately (Supplementary Data 7). When determining the concordance between eQTLGen's bulk whole-blood eQTLs and those identified in our study, we observed that the concordance was high in general despite the compositional differences between whole blood and the PBMCs or cells in this study that were pathogen-stimulated. As expected, we obtained the highest concordance (95.5%) with eQTLGen when compared to our bulk-like unstimulated PBMC scRNA-seq data, i.e., taking the average gene expression across all cells from one individual in the untreated condition (Fig. 3a). We then saw only a minor drop to 94.7% concordance when comparing eQTLs from eQTLGen with our pathogen-stimulated (24 h CA) bulk-like scRNA-seq dataset (Fig. 3b) and a further decrease to 92.6% when compared to our pathogen-stimulated and cell-type-specific scRNA-seq dataset (24 h CA in monocytes) (Fig. 3c). Finally, to verify that our initial selection of eQTLGen lead-eSNPs did not confound our conclusions, we also compared the output of a genome-wide cis-eQTL discovery (Supplementary Data 8) in pathogen-stimulated and cell-type-specific scRNA-seq data (24 h CA in monocytes) with eQTLGen. In this analysis, the concordance decreased a little bit further to 87.4% (Fig. 3d). Although up to 19.6% of the eQTLs were only detected in the genome-wide discovery (and not in the eQTLGen lead-eSNP-confined cis-eQTL discovery), these unique cis-eQTL gene sets were not enriched for specific biological pathways. Altogether, this indicated that the eQTLGen lead-eSNP confinement only had a minimal impact on our observations and confirmed our initial assumption that the majority of context-specific eQTLs identified by our current study can already be detected in very large bulk RNA-seq datasets. However, we still require single-cell data to pinpoint their relevant context. As the eQTLGen lead-eSNP cis-eQTL analysis identified 1.5× more eQTLs, while showing no clear bias towards common eQTLs rather than cell-type-specific or context-dependent eQTLs (Supplementary Fig. 4a), we continued our analysis with these results.

The CD4+ T cells revealed the most eQTL effects, followed by the monocytes and CD8+ T cells (Supplementary Fig. 4a). The cell types with the lowest frequencies, the DCs and B cells, also showed the lowest number of eQTLs (Supplementary Fig. 4b). This large difference in the number of identified eQTLs per cell type is, at least in part, explained by the difference in power, given the number of cells of each cell type (Supplementary Fig. 1g). When overlapping the identified eQTL genes in the major cell types with each of their two subcell types (overall stimulation-timepoint combinations combined), we observed that the majority of eQTL genes identified in the subcell types were already detected in the corresponding major cell type (Supplementary Fig. 4c). Nevertheless, 4.6% (for the NKdim) up to 24.5% (for the pDCs) additional eQTL genes were uniquely identified in such a subcell type.

In addition to differences between cell types, we also observed differences between stimulation–timepoint combinations (Supplementary Fig. 4d). However, direct comparisons of the number of eQTLs between conditions within the same cell type were complicated because the number of included individuals varied among the stimulation–timepoint combinations as a result of QC dropouts (UT: 104 individuals, 3 h CA: 120 individuals, 3 h MTB: 104 individuals, 3 h PA: 112 individuals, 24 h CA: 119 individuals, 24 h MTB: 112 individuals, 24 h PA: 111 individuals). Most interestingly, when comparing the effect of pathogen stimulation on the number of identified eQTLs between cell types, we

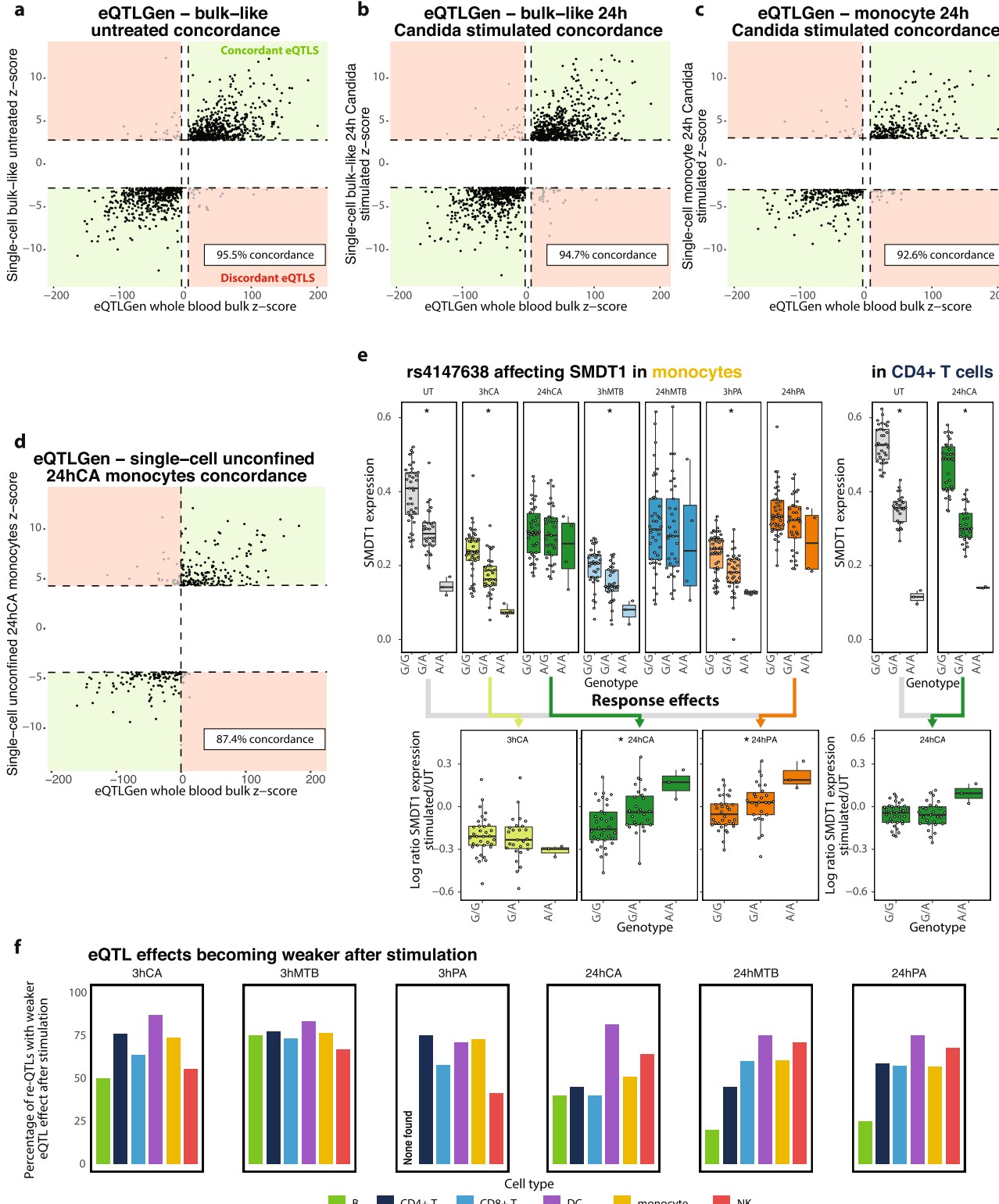

observed an inverse correlation with the responsiveness of that cell type to pathogen stimulation (Supplementary Fig. 4e). For example, the myeloid cells showed the largest DE response upon pathogen stimulation (Fig. 2a) but a consistent reduction in the number of eQTLs identified after stimulation (Supplementary Fig. 4a). In contrast, the lymphoid cells showed a much smaller DE response upon pathogen stimulation (Fig. 2a) but an increase in the number of eQTLs identified after stimulation, in about half of the conditions (Supplementary Fig. 4a). This could indicate

that, for at least a subset of the genes, the influence of genetics on gene expression may become more restricted when cells have to orchestrate a response to an environmental stimulus[38].

To identify eQTLs for which the strength of the eQTL effect was affected by pathogen stimulation, we performed a response-QTL (re-QTL) analysis[39]. We systematically looked for re-QTLs in all major cell types and stimulation conditions compared to the untreated condition (Supplementary Data 9). Most re-QTLs were specific to a particular timepoint or cell type, but less so to a

**Fig. 3 eQTLs and re-QTLs upon pathogen stimulation.** Concordance between the eQTLs identified in 31,684 bulk whole-blood samples of the eQTLGen consortium and: **a** those identified in our eQTLGen lead-eSNP discovery of bulk-like unstimulated PBMC scRNA-seq data, (**b**) those identified in our eQTLGen lead-eSNP discovery of bulk-like 24 h *C. albicans* (CA)-stimulated PBMC scRNA-seq data, (**c**) those identified in our eQTLGen lead-eSNP discovery of monocyte 24 h CA-stimulated PBMC scRNA-seq data, and (**d**) those identified in our genome-wide eQTL discovery of monocyte 24 h CA-stimulated PBMC scRNA-seq data. **e** Boxplots showing the effect of the rs4147638 genotype on SMDT1 expression in the untreated (UT) condition and each of the six stimulation–timepoint combinations in the monocytes (left) or for the UT and 24 h CA condition in the CD4+ T cells (right). Boxplots show median, first and third quartiles, and 1.5× the interquartile range, and each dot represents the average expression of all cells per cell type and individual. Stars indicate a significant effect (FDR < 0.001). The log ratio of SMDT1 expression in the UT cells vs a specific stimulation-timepoint combination is shown in the bottom. Colored arrows indicate which specific stimulation–timepoint combination was selected for the corresponding re-QTL boxplot. **f** The proportion of re-QTLs of which the eQTL effect became weaker after stimulation, split per cell type and stimulation–timepoint combination. eQTL summary statistics for eQTLGen-confined analysis, genome-wide analysis and response-QTL analysis can be found in Supplementary Data 7, Supplementary Data 8, and Supplementary Data 9, respectively. The number of individuals and cells included in each analysis can be found in the Source Data file.

particular pathogen (Fig. 3e). We observed that most re-QTLs were in the monocytes for each of the stimulation–timepoint combinations (Supplementary Fig. 4a), likely the direct consequence of the combination of a high number of DE genes upon stimulation (Fig. 2a) and the relatively high number of monocytes per individual (Supplementary Fig. 1g). We also observed that most re-QTLs describe eQTL effects that became weaker after stimulation (Fig. 3f). Of those eQTL effects that became stronger after stimulation, 26.3% on average showed a significant effect that was already present in the unstimulated samples, whereas those effects were not yet present for 63.7%. Moreover, we observed clear enrichment of DE genes within the set of eQTL and re-QTL genes, but this enrichment was not consistently greater for re-QTL in comparison to eQTL genes (Supplementary Fig. 4f).

Finally, when linking the eSNP loci identified in each of the major cell types to GWAS output of immune-mediated diseases (see Methods), we observed strong genomic inflation across all conditions (Supplementary Data 10). This genomic inflation increased further for the re-QTLs (in monocytes across all immune-mediated GWASes: $p = 0.024$) (Supplementary Data 10). These findings confirmed previous studies showing that stimulation-responsive eQTL effects provide additional explanation of immune-mediated disease risk over baseline eQTLs[4,40]. Additionally, it has been shown that the effect size of GWAS-associated SNPs becomes larger in the disease-relevant context (e.g., immune-mediated disease patients as opposed to the healthy controls)[41]. Therefore, also the power to detect these disease-associated effects will be larger in the disease-relevant context.

In summary, we observed that 20.9% of our eQTL genes that were identified in the major cell types were influenced by a combination of genetics and environment (Supplementary Data 7, Supplementary Data 9). We expect this percentage is underestimated, as the power to detect re-QTLs is inherently lower than that of eQTLs, and exposure to additional environmental stimuli may reveal additional context-dependency. Altogether, our findings indicate that in addition to cell-type specificity, context-dependency is also a major driver of genetic regulation of gene expression and provides an additional explanation of disease risk.

**Pathogen stimulation induces widespread context-specific gene regulation.** We have previously shown that genetics can influence the co-expression relationship between genes and that scRNA-seq data are uniquely suitable to do so by taking the individual cells per cell type per donor as observations over which the individual-specific co-expression is calculated[12]. In contrast, bulk RNA-seq data usually contain a single measurement per donor, and therefore, co-expression in bulk data cannot be calculated at the individual level. As a consequence, the co-expression between two genes as calculated from bulk RNA-seq data may be different

from the true individual-specific co-expression relationship as extracted from scRNA-seq data (due to Simpson's paradox[42]).

In addition, studies that compared co-expression in healthy versus disease states have indicated that environmental conditions may also impact this gene–gene interactions[43]. Here, we took the next step by determining whether and how the combination of genetics and environment may affect how genes are interacting with one another by performing co-expression QTL analysis, i.e., a SNP genotype affecting the co-expression relationship of a gene pair. For this purpose, we selected a subset of 49 SNP–gene combinations that we then tested against up to 5772 genes. To enrich for SNP–gene combinations in which we expect an interaction with the environment, we selected these based on: (1) the gene being DE and (2) the SNP–gene combination being a re-QTL in at least one of the stimulation–timepoint combinations; (3) the gene being expressed in at least 50% of the individuals (in each 10x chemistry). For this analysis, we focused solely on the monocytes because this was the cell type that showed a strong response to pathogen stimulations and for which we had sufficient cells per individual (i.e., hundreds) to perform a robust co-expression QTL analysis[44]. By making this pre-selection of 49 SNP–gene combinations, we could reduce the multiple testing burden from over $10^{14}$ in a genome-wide analysis to fewer than 283,000 tests.

Across the unstimulated condition and each of the six stimulation–timepoint combinations, we found at least one co-expression QTL for 35 SNP–gene combinations and more than 100 co-expression QTLs in at least one condition for 9 SNP–gene combinations. For each of these 9 SNP–gene combinations with a high number of co-expression QTLs, we observed an interaction between genotype and stimulation condition (Fig. 4a, Supplementary Data 11). One of these co-expression QTLs described an interaction between *RPS26* and rs1131017, which was an effect in high linkage disequilibrium(LD) with one we had identified as a co-expression QTL in CD4+ T cells in our previous study (rs7297175, $R^2 = 0.92$)[12]. rs1131017 was previously associated with rheumatoid arthritis[45] ($p = 1.3 \times 10^{-8}$) and is in high LD with a type I diabetes GWAS SNP[46] (rs11171739, $R^2 = 0.94$). For this *RPS26*-rs1131017 SNP combination, we found 1701 co-expression QTLs in the unstimulated condition. Of the 106 *RPS26* co-expression QTLs that we had previously identified in CD4+ T cells[12], 72 (67.9%) were also found in the unstimulated monocytes in our current study (91.7% with the same allelic direction) (Supplementary Fig. 5a). Any discrepancy between these two cell types might be the consequence of distinct regulatory mechanisms that are active in those cell types. Next, looking at the effect of stimulation, the number and strength of the detected *RPS26* co-expression QTLs reduced greatly after stimulation and were related to the duration of stimulation: on average we observed 459 co-expression QTLs after 3 h

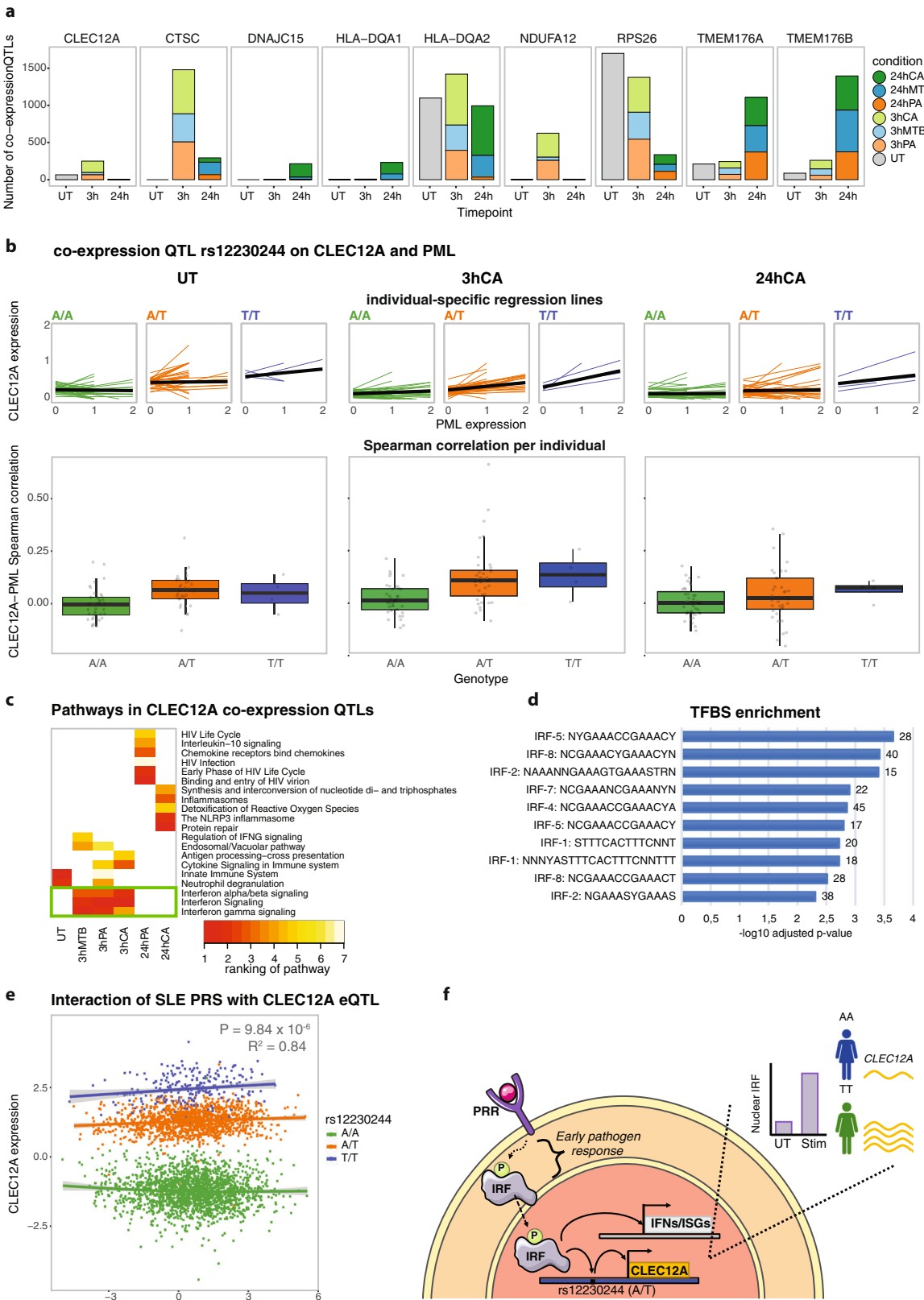

**a**

**b** co-expression QTL rs12230244 on CLEC12A and PML

**c** Pathways in CLEC12A co-expression QTLs

**d** TFBS enrichment

**e** Interaction of SLE PRS with CLEC12A eQTL

**f**

stimulation and 112 after 24 h stimulation (Fig. 4a, Supplementary Fig. 5b).

We also observed this general decrease in the strength and number of co-expression QTLs with increasing duration of pathogen stimulation for the *HLA-DQA2* co-expression QTLs, but not for any of the other 7 SNP–gene combinations (Fig. 4a).

These other seven co-expression QTL effects increased in strength and number upon stimulation (Fig. 4a, Supplementary Data 11). Interestingly, for some of these co-expression QTL genes, we observed the most prominent increase at 3 h stimulation (i.e., *CLEC12A, CTSC,* and *NDUFA12*), whereas others were more prominent at 24 h stimulation (i.e., *TMEM176A/B, DNAJC15,*

**Fig. 4 Interferon regulatory factor affects *CLEC12A* co-expression QTLs upon 3 h pathogen stimulation in monocytes. a** Number of co-expression QTLs identified in each of the stimulation–timepoint combinations for those co-expression QTLs with over 100 co-expression QTLs in at least one condition. The 3 h and 24 h timepoint are colored by pathogen stimulation (green: *C. albicans* (CA), blue: *M. tuberculosis* (MTB), orange: *P. aeruginosa* (PA). Co-expression QTL summary statistics can be found in Supplementary Data 11. **b** The lines in the top plots show co-expression between *CLEC12A* and *PML* (most significant co-expression QTL across the 3 h stimulation conditions) for individual cells in the untreated (left), 3 h CA (middle) and 24 h CA (right) condition. In these plots, individual-specific regression lines are shown, split by genotype. The average genotype-specific regression lines are shown in black. The bottom boxplots depict Spearman's rank correlation between *CLEC12A* and *PML* expression, stratified by SNP rs12230244 genotype in the monocytes per individual, in the untreated (left), 3 h CA (middle) and 24 h CA (right) stimulated cells (the V2 chemistry data is plotted). Each data point shows a single individual. Boxplots show median, first and third quartiles, and 1.5× the interquartile range. **c** Heatmap of the top-5 enriched pathways within the co-expressed *CLEC12A* co-eQTL genes per stimulation–timepoint combination. Per combination, pathways are ranked based on significance. White indicates that the pathway was not found to be enriched in that specific stimulation–timepoint combination. The green box highlights pathways that are associated with all 3 h stimulation conditions. **d** Top 10 enriched putative transcription factor binding sites within the *CLEC12A* co-expression QTL genes that: (1) showed a more positive strength of the co-expression relationship in individuals with the TT as opposed to the AA genotype and (2) were identified in the 3 h stimulated (outer join) monocytes using the TRANSFAC database. Enrichment of putative transcription factor binding sites was defined using a g:SCS multiple testing correction method, applying a significance threshold of 0.05. **e** Co-expression QTL analysis for *CLEC12A*-SNP rs12230244 against the SLE polygenic risk score (PRS) (calculated using those SLE GWAS SNPs with a *P*-value threshold of $<5 \times 10^{-8}$) using whole-blood bulk expression data from 3553 individuals (BIOS consortium). A one-tailed F-test (coefficient = 0.04, standard error = 0.01, f-value = 19.60, p-value = $9.84 \times 10^{-6}$, $R^2$ = 0.84) was used to determine whether the distribution of the squared residuals with the SLE PRS as interaction term was significantly smaller than without. **f** Proposed mechanism of action of *CLEC12A* co-expression QTLs. When pathogen-associated molecular patterns bind to a pattern recognition receptor (PRR), a signaling cascade is initiated that eventually results in phosphorylation of interferon regulatory factors (IRFs). Phosphorylated IRF then translocates to the nucleus, where it binds to specific DNA motifs such as IFN-stimulated response elements. This can then activate transcription of IFNs and IFN-stimulated genes (ISGs). Additionally, IRF is expected to bind to a region containing SNP rs12230244 (or any another SNP in high LD), thereby regulating *CLEC12A* expression. In this case, depending on the SNP genotype, the IRF binding and activation of *CLEC12A* expression is expected to be stronger (TT genotype) or weaker (AA genotype). Many of the identified *CLEC12A* co-expression QTL genes are involved in the IFN pathway (see panel **b**). This has to be the result of a common upstream factor (i.e., IRF) of *CLEC12A* transcription that can also activate IFNs and ISGs, but cannot be the result of a downstream regulator because this would have led to trans-eQTL effects for the same SNP rs12230244 (which we do not observe). The number of individuals and cells included in each analysis can be found in the Source Data file.

and *HLADQA1*). The observation of different numbers of co-expression QTLs for a specific gene over the seven stimulation–timepoint combinations was not fully explained by the expression level of that gene. Beyond this variation over the timepoints, we also observed clear differences between the various pathogen stimulations. At gene level, there was little overlap between the co-expressed gene sets between the different pathogen stimulations (Supplementary Data 11), whereas this overlap was much larger at pathway level (Supplementary Fig. 5). The low gene-level overlap is likely a consequence of power and is something that will be largely overcome in the near future with the increase in the number of cells per dataset[47,48]. Together, these results indicate that specific environmental conditions can fulfill the requirements needed for a specific co-expression QTL interaction to occur.

Previously, re-QTL analyses in cells exposed to highly specific stimuli were used to disentangle the environmental conditions that underlie specific genetic regulation of gene expression[4,16]. However, this has the disadvantage that either many highly specific stimuli have to be applied, or, in the case of applying broad stimuli, the exact environmental context relevant for the interaction remains vague. Here, we propose using co-expression QTL analysis upon stimulation with a few broad stimuli to gain this detailed insight in a more unbiased way, without the need to apply many highly specific stimuli. As a first example of how co-expression QTL analysis can help us understand the underlying mechanisms of gene regulation, we focused on the *CLEC12A* co-expression QTLs affected by SNP rs12230244, which were most prominent at 3 h of pathogen stimulation (Fig. 4a, b). *CLEC12A*, also known as *MICL*, encodes for an inhibitory C-type lectin-like receptor and is mostly expressed in myeloid cells such as monocytes and DCs. CLEC12A signaling can be activated by the binding of uric acid crystals, which are the byproduct of nucleic acids that can be released from damaged or dying cells[49,50]. Activation of CLEC12A signaling can result in inhibition of the activating C-type lectin receptors and can prevent hyperinflammation during necrosis[51].

To identify the potential causal factor underlying the *CLEC12A* co-expression QTL, we performed a pathway analysis on the associated co-expressed gene set of each of the stimulation–timepoint combinations. We hypothesized that co-expressed genes linked to the same co-expression QTL mostly describe the same (or only a few) biological processes that are driven by a single (or a few) causal factors being directly involved, and that most of these co-expressed genes are themselves just a consequence of being highly co-expressed with the causal factor. An important category of causal factors is transcription factors. However, average expression levels of transcription factors are generally low and, particularly in dynamic situations such as a pathogen response, mRNA levels might not correlate well with the nuclear protein expression levels (i.e., the functional proportion)[52,53]. Consequently, it can be difficult to define the direct causal factor solely using co-expression QTL analysis. Nevertheless, we expected that by taking a pathway-level view, the downstream genes of transcription factors would have a high correlation with the functional protein level of the transcription factor and would be more easily picked up than a single gene.

The pathway analysis of the *CLEC12A* genotype-dependent co-expressed gene set after 3 h stimulation (Benjamini-Hochberg (BH) corrected $p = 2.9 \times 10^{-5}$, $8.7 \times 10^{-7}$, and $4.3 \times 10^{-4}$ for 3 h CA, 3 h PA, and 3 h MTB, respectively), but not in the untreated or 24 h stimulation conditions, revealed enrichment of the IFN pathway (Fig. 4c). This result hinted that a component within or regulating the IFN pathway could be the causal factor that is regulating the different *CLEC12A* co-expression responses per genotype after 3 h stimulation. To provide additional support for this hypothesis, we performed a functional enrichment analysis for putative transcription factor binding sites (TFBSs) (TRANS-FAC database[54]) on the *CLEC12A* genotype-dependent co-expressed genes upon 3 h stimulation. We divided this gene set into a subset in which individuals with the TT as opposed to the AA genotype showed a more positive rather than a more negative co-expression relationship between *CLEC12A* and its co-

expressed genes, as potentially different mechanisms could be underlying these gene sets. This analysis revealed no enriched TFBSs in the negative-strength gene set, but a clear enrichment of various IFN regulatory factors (IRFs) in the positive-strength gene set, including *IRF1, 2, 4, 5, 7,* and *8* (Fig. 4d). Additionally, when overlapping the *CLEC12A* co-expression QTL SNP rs12230244 and its accompanying (near-)perfect LD SNPs with putative TFBSs[55], we observed several transcription factors that may bind to the genomic location of these SNPs. Most notably, the predicted binding site of *IRF1* was shown to be enriched in the genomic location of two SNPs that are in near-perfect LD with the *CLEC12A* co-expression QTL SNP: rs999185 ($R^2 = 0.9943$) and rs57106602 ($R^2 = 0.9$).

Finally, we used two external datasets and slightly different approaches to further strengthen our hypothesis that IFN activity is regulating the *CLEC12A* co-expression QTL effects. First, we used the BIOS consortium bulk RNA-seq dataset containing whole-blood data from 3553 individuals[56]. For each of those individuals, we calculated a polygenic risk score (PRS) for the autoimmune disease systemic lupus erythematosus (SLE), a disease characterized by increased type I IFN activity[57–60]. We reasoned that the genetic risk captured by the SLE PRS could be used as a proxy for IFN activity. Consequently, the difference in the co-expression relationship between the SLE PRS and the *CLEC12A* per rs12230244 genotype indicated the involvement of IFN signaling in this interaction (Fig. 4e). Second, we used an independent scRNA-seq dataset generated in 68 healthy controls and 117 SLE patients of European (EUR) and East Asian (EAS) origin[61]. We reasoned that since IFN activity is characteristic for SLE[57–60], SLE patients would mimic the 3 h pathogen-stimulation state in which high IFN activity seems to drive the observed *CLEC12A* co-expression QTL effects. We also reasoned that the healthy controls would mimic the untreated cells in our study and therefore show fewer *CLEC12A* co-expression QTL effects driven by IFN activity. To define whether the SLE patients mimicked the results as observed after 3 h pathogen stimulation, we performed a co-expression QTL analysis for *CLEC12A* and SNP rs12230244 in the monocytes of SLE patients and healthy controls (Supplementary Data 12). Pathway analysis on the *CLEC12A* co-expression QTL genes revealed stronger enrichment for the IFN pathway in the SLE patients (FDR = $1.965 \times 10^{-7}$) compared to the healthy controls (FDR = $1.203 \times 10^{-3}$), again supporting that this pathway is involved in the regulation of *CLEC12A* through the locus with the rs12230244 SNP.

As a second example, we applied a similar strategy to learn the underlying regulatory mechanism by which the co-expression QTLs identified for SNP rs6945636 affect the heat-shock protein response gene *ZFAND2A*. The heat-shock protein response is a pathway that, among others, can be activated by bacterial and viral infections[62]. We selected this specific SNP–gene combination for further analysis because the co-expression QTLs identified were both pathogen- and timepoint-specific (only at the 24 h timepoint, 96% of the genes being detected in CA only). Pathway analysis of the co-expressed genes revealed 'Intracellular pH reduction' (GO:0051452) as the top associated biological process (adjusted $p = 3.8 \times 10^{-4}$). Interestingly, *HSF1*, a known regulator of *ZFAND2A*[63] was shown to be pH sensitive in yeast[64]. Moreover, the *ZFAND2A*-associated co-expression QTL SNP rs6945636 was in almost perfect LD with previously identified *HSF1* binding sites in K562 cells ($R^2 = 0.99$, rs715188378; $R^2 = 0.99$, rs79849558; $R^2 = 0.99$, rs11767061, retrieved from dbSNP release 153)[65]. Together, this indicates that CA-induced pH regulation activated *HSF1*, which in turn bound with stronger (TT genotype) or weaker (AA genotype) strength to rs6945636 SNP locus, and thereby strongly or weakly activated *ZFAND2A*, respectively.

These two examples provide clear use cases for how co-expression QTL analysis can be applied to gain detailed insights into the underlying context of gene expression regulation. For example, in the case of *CLEC12A*, without co-expression QTL analysis we could only reveal that *CLEC12A* is a re-QTL regulated by a factor active 3 h downstream of pathogen stimulation (Supplementary Data 9). In contrast, using co-expression QTL analysis, we were directed to the causal regulatory factor for this re-QTL. This enabled follow-up analyses that gathered solid evidence for the following mechanism of action through which the rs12230244 SNP locus affects *CLEC12A* expression specifically upon 3 h pathogen stimulation: (1) pathogen-associated molecular patterns bind to a pattern recognition receptor (PRR) and initiate a signaling cascade that eventually results in phosphorylation of interferon regulatory factors (IRFs), (2) phosphorylated IRF then translocates to the nucleus where it binds to specific DNA motifs such as IFN-stimulated response elements, and 3. this can then activate transcription of IFNs and IFN-stimulated genes (ISGs). Additionally, IRF is expected to bind to a region containing SNP rs12230244 (or any other SNP in high LD), thereby regulating *CLEC12A* expression. In this case, depending on the SNP genotype, the IRF binding and induction of *CLEC12A* expression is expected to be stronger (TT genotype) or weaker (AA genotype) (Fig. 4f).

Interestingly, we identified a number of (near-)genome-wide significant PheWAS traits related to immune cell composition and size to be associated with these two co-expression QTL loci (extracted from 452,264 White British individuals of the UK Biobank[66]): platelet counts ($p = 2.1 \times 10^{-8}$), monocyte percentage ($p = 1.8 \times 10^{-5}$) and eosinophil counts ($p = 6.4 \times 10^{-5}$) for *CLEC12A* and mean corpuscular volume ($p = 1.5 \times 10^{-14}$) and mean sphered cell volume ($p = 2.6 \times 10^{-9}$) for *ZFAND2A*. However, no direct association was found for any of the immune-related GWAS tested (SLE[67], inflammatory bowel disease[68], celiac disease[69], rheumatoid arthritis[45], multiple sclerosis[70], type I diabetes mellitus[71], and candidemia[72], which had 10–2541-fold smaller sample sizes than the PheWAS. This overlap with immune-related PheWAS traits indicates the relevance of these SNPs for immune function.

Moreover, looking at the function of the affected genes, we also expect immunological consequences of the identified co-expression QTLs. For example, previously *CLEC12A* was shown to act as an early adaptor molecule for antibacterial autophagy, and in mice, complete knockout of Clec12a resulted in higher susceptibility to Salmonella infection[73]. Additionally, *CLEC12A* is known to contribute to the pathogenesis of rheumatoid arthritis. For example, upon collagen-induced arthritis, *CLEC12A* knock-down mice show increased joint inflammation[74], and in monocytes of early rheumatoid arthritis patients reduced expression of *CLEC12A* correlated with more severe disease 6 months later[75]. Together, this suggests that individuals with the AA allele on rs12230244 may be at increased risk of bacterial infection and of developing joint inflammation, acting through reduced induction of *CLEC12A* expression when exposed to pathogens or other factors inducing IFN signaling.

Summarized, using co-expression QTL analysis, we can now dissect the underlying mechanism by which such an effect is regulated. This information will help explain the downstream consequences on immune function, and potentially enable new routes for medical intervention.

## Discussion

GWAS studies have provided important insights into the genetic architecture of phenotypic traits and diseases[1]. However, the exact mechanisms by which genetic variation leads to these traits

or diseases largely remain a black box. To uncover these mechanisms, various approaches have been successfully applied, for example coupling the trait-associated risk factor to the nearest positional gene[76], downstream gene expression[56], or gene regulation[12]. Nevertheless, a large knowledge gap remains that may, in part, be filled by taking into consideration the context in which the genetic variant can lead to disease[7,16,17].

To uncover the interplay between genetics and cellular and environmental context, we single-cell RNA-sequenced PBMCs from 120 individuals from Lifelines, a large population-based cohort from the Northern Netherlands, that had been exposed to various pathogens or left untreated. Subsequent DE, eQTL, and co-expression QTL analysis revealed that there are widespread interactions between an individual's genetics and the cellular and environmental context, both at the level of gene expression and in its regulation. We identified hundreds of eQTLs in the individual cell types and upon pathogen stimulation and observed strong context-specificity for 25.7% of the identified co-expression QTLs. In general, we observe more interactions between genetics and cell-type-specific context, as opposed to context induced by pathogen stimulation. However, some of these differences may have been the result of differences in detection power. Contrary to expectations, in the cell types with the strongest response to pathogen stimulation (i.e., the myeloid cells), the total number of eQTLs were reduced after stimulation. Moreover, in all cell types, we observed that eQTLs more often became weaker rather than stronger after stimulation and that neither category of eQTL genes was associated with a specific pathway. In contrast, for the co-expression QTL genes, the number of co-expressed genes more often increased upon pathogen stimulation. However, this might in part have been the result of our selection, i.e., choosing re-QTLs in monocytes as the starting point for the co-expression QTL mapping. Moreover, we observed genomic inflation of eQTLs that further increased when focusing solely on the re-QTLs. Altogether, these observations indicate that context, here the pathogen-stimulation condition, is an important contributor that affects the association between SNPs and gene expression or co-expression, and that taking this context in consideration further improves our understanding of disease risk.

A major advantage of co-expression QTL analysis, as opposed to re-QTL analysis, is that we do not require many highly specific stimuli to disentangle the mechanisms that underlie the context-specificity of the genetic regulation. Instead, in this study, we have shown that, after applying a broad stimulation (i.e., whole-pathogen stimulation), a wide range of contexts are activated, and that, through subsequent co-expression QTL analysis, the specific context and mechanism of action could be uncovered. For example, we revealed that an interferon-regulated transcription factor was affecting the SNP rs12230244−dependent downstream activation of *CLEC12A*. Additionally, we showed how pH-dependent regulation of the heat-shock protein response transcription factor *HSF1* affected the SNP rs6945636−dependent downstream activation of *ZFAND2A*. Even though the causal SNP cannot be conclusively determined using co-expression QTL analysis, understanding the underlying mechanism can help to further fine-map the genetic signal. These examples clearly show the potential of the technology and provide an outlook into where the field will be moving as more population-scale scRNA-seq datasets become available. We foresee that newly developed methodology, such as inCITE-seq[53] and NEAT-seq[77], combining measurements of multiple omics layers from the same cell, including RNA and nuclear protein levels (which allows measuring active transcription factors levels), will further enhance the interpretability of the identified co-expression QTLs in the future.

Importantly, this study was conducted on European individuals with a white background. Although we do not expect general conclusions to be different in other populations, it may be that the upstream regulators or downstream consequences of some of the specific genetic variants act differently across populations. Moreover, as the infection history with the three pathogens understudy is unknown for the individuals included in our study, there is a small chance that this may have introduced additional noise or confounding in our analyses.

In the last few years, scRNA-seq has become a mature, high-throughput technology[8,9]. This has led to several initiatives aiming to study population genetics at single-cell resolution, such as the sc-eQTLGen consortium[47] and others[78]. Such efforts bring together many single-cell eQTL studies, conducted on individuals from different ethnicities and exposed to different environments or diseases. This will not only increase the power to detect eQTLs and co-expression QTLs, it will also further extend our findings to additional contexts and enable genome-wide cell-type and context-specific *trans*-eQTL mapping. Moreover, instead of linking individual genetic variants, linking polygenic risk scores to cell-type-specific gene expression (i.e., eQTS analysis[37]) may provide a more disease-focused insight into how the combination of disease-associated variants together contributes to changes in gene expression levels. By integrating GWAS signals, PRSes, and context-specific QTL information, we expect that these efforts can drive major leaps forward in disease understanding and precision medicine[79].

## Methods

### Ethics approval and informed consent
The LifeLines DEEP study was approved by the ethics committee of the University Medical Centre Groningen, document number METC UMCG LLDEEP: M12.113965. All participants signed informed consent from prior to study enrollment. All procedures performed in studies involving human participants were in accordance with the ethical standards of the institutional and/or national research committee and with the 1964 Helsinki declaration and its later amendments or comparable ethical standards.

### PBMC collection and stimulations
Whole blood from 120 European white background individuals of the northern Netherlands population cohort Lifelines Deep[80] was drawn into EDTA-vacutainers (BD). PBMCs were isolated and maintained, as previously described[12]. In short, PBMCs were isolated using Cell Preparation Tubes with sodium heparin (BD) and were cryopreserved until use in RPMI1640 containing 40% FCS and 10% DMSO. After thawing and a 1 h resting period, unstimulated cells were washed twice in a medium supplemented with 0.04% BSA and directly processed for scRNA-seq. In contrast, for stimulation experiments, $5 \times 10^5$ cells were seeded in a nucleon sphere 96-well round-bottom plate in 200 μl RPMI1640 supplemented with 50 μg/mL gentamicin, 2 mM L-glutamine, and 1 mM pyruvate. Then, in vitro stimulations were applied for either 3 h or 24 h using $1 \times 10^6$ CFU/ml heat-killed *C. albicans blastoconidia* (strain ATCC MYA-3573, UC 820), 50 μg/ml heat-killed *M. tuberculosis* (strain H37Ra, Invivogen) or $1 \times 10^7$ heat-killed *P. aeruginosa* (Invivogen) while incubating the cells at 37 °C in a 5% $CO_2$ incubator. After stimulations, cells were washed twice in a medium supplemented with 0.04% BSA. Cells were then counted using a hemocytometer, and cell viability was assessed by Trypan Blue.

### Single-cell library preparation and sequencing
105 sample pools were prepared, each aimed to yield 1400 cells/individual from eight individuals (11,200 cells). In general, pools contained a mixture of both sexes and two different stimulation conditions. Each sample pool was loaded into a different lane of a 10× chip (Single Cell A Chip Kit for v2 or Single Cell B Chip Kit for v3 reagents). The 10x Chromium controller (10x Genomics), in combination with v2 (72 libraries) or v3 (33 libraries) reagents, was used to capture the single cells and generate sequencing libraries, according to the manufacturer's instructions (document CG00052 and CG000183 for v2 and v3, respectively) and as previously described[12]. Sequencing was performed with a 150 bp paired-end kit using a custom program (V2: 27-9-0-150, V3: 28-8-0-150) on the Illumina NovaSeq 6000 at BGI (Hong Kong).

### scRNA-seq alignment, preprocessing, and QC
CellRanger v3.0.2 was used with default parameters to demultiplex, generate FASTQ files, align reads to the hg19 reference genome, filter both cell- and unique molecular identifier (UMI) barcodes, and count gene expression per cell. To assign cells to one of the eight individuals in a lane, Demuxlet was used[11]. The genotype information used by Demuxlet was previously generated as described in Tigchelaar et al.[80] and was phased with Eagle v2.322 using the HRC reference panel and the Michigan Imputation Server. Only exonic variants with a MAF of at least 0.02 were used for demultiplexing.

Subsequently, Souporcell v1.0[29] was used to remove doublets coming from different individuals, by looking for the different genotypes within a single-cell assignment. We limited the SNP calling to positions that were also used for demultiplexing.

Version 3.1 of the Seurat[81] package was used for further quality control and processing. Due to mRNA capture differences between the v2 and v3 chemistries, a version-chemistry-specific maximum mitochondrial gene content percentage of 8% and 15% was used, respectively. Cells with less than 200 detected genes were discarded, as well as cells with more than 9 UMIs mapping to the hemoglobin subunit beta (HBB) gene (representing red blood cells), and other low-quality cells (i.e., clusters of cells with a low number of expressed genes and a relatively high mitochondrial content, or missed, likely same-individual, doublets) (Supplementary Data 3).

For annotating the cell types, we first log-normalized the count matrices for each of the seven timepoint-stimulation conditions and two chemistries separately using Seurat's LogNormalize function (scale.factor = 10,000)[81]. The log-normalized count matrices of the unstimulated data were then integrated separately for each of the three pathogen stimulations. For this, we used the first 30 dimensions from a Canonical Correlation Analysis to identify integration anchors in Seurat's FindIntegrationAnchors function. These anchors were then used for integration using Seurat's IntegrateData function[81]. We performed principal component analysis (PCA) and selected the first 30 principal components to identify the cell clusters using k-nearest neighbor clustering and visualized this in UMAP space (using the default settings). Cell types were assigned to each cluster based on marker gene expression, resulting in a set of six major cell types and ten subcell types (Supplementary Fig. 1B, Supplementary Data 4). A small fraction of the cells could not be classified at higher resolution, and therefore, where omitted from the subcell type analyses (Source Data file). For each version chemistry, gene expression counts were then SCT normalized using Seurat's SCTransform function, and cell-type labels obtained from the integrated data were transferred to non-integrated data (Fig. 1A), to preserve the stimulation response at the gene expression level.

### Differential expression: mapping, pathway enrichment, and module scoring.
For each pathogen-timepoint combination, major and subcell type and 10x chemistry, differential expression (DE) analyses were performed between the pathogen-stimulated and the untreated condition using the MAST implementation of Seurat[30]. Testing was limited to genes with a log-fold change (LFC) > 0.1 and with expression in at least 10% of the cells. We used MetaVolcanoR[82] to perform a meta-analysis for each cell type, taking the results of the v2 and v3 chemistries as inputs for Fishers Combined Probability Test[83]. Significance was determined by taking a Bonferroni-corrected p-value of <0.05 within the meta-analysis. When an analysis could only be performed in one version chemistry, only that output is reported.

Per cell type, the resulting DE gene set was split into up- and downregulated genes after stimulation, which was then used as input for a pathway enrichment analysis with ToppFun, selecting the REACTOME database[84]. To calculate statistical significance, the probability density function was used, selecting those pathways that had a BH-corrected p-value < 0.05.

For the major cell types, the enriched pathways were visualized by calculating the LFC in average gene expression in all pathogen-timepoint conditions compared to the untreated condition and clustered these results using hierarchical clustering with the complete linkage method. For the subcell types we made the comparisons only within the subtypes that fall within the same major cell type (Supplementary Data 4). We visualized up to 10 enriched pathways that showed the largest difference between the two subtypes (within the same major cell type) and ordered these pathways by the difference in log10 transformed significance between the cell types. The fraction of genes that were found to be differentially expressed versus the total annotated genes in the gene sets, was determined by dividing the differentially expressed genes found for each gene set, by the total number of genes of a gene set.

Calculation of pathway activity was done using the module score function of Seurat[85], by calculating, per cell, the combined activity of a specific gene set annotated to be part of a pathway in the REACTOME database. This score was then averaged per donor for each condition and cell type.

### eQTL and re-QTLs: mapping and GWAS enrichment.
The mapping of eQTLs was performed in a bulk-like and cell-type-specific manner. We limited the analysis to the top independent effects identified in the eQTLGen meta-analysis on 31,684 individuals, resulting in the testing of 16,987 possible SNP-gene pair combinations[56]. These SNP-gene combinations identified by eQTLGen were the result of genome-wide cis-eQTL mapping of SNPs within a 100 kb distance to the gene midpoint, MAF > 0.1, call rate >0.95, and Hardy-Weinberg equilibrium p-value > 0.001. These 16,987 SNP-gene pairs were then further filtered to only include SNPs with a minor allele frequency (MAF) > 0.1 or genes that were expressed in the least three cells in our single-cell data. Filtering of SNP-gene combinations and mapping of eQTLs were done separately for each cell type and reagent version chemistry using the averaged, normalized gene expression values per individual, cell type, and stimulation-timepoint combination. This was followed by a sample-weighted meta-analysis[86] over the v2 and v3 chemistry outputs per cell type and stimulation-timepoint combination.

When an analysis could only be performed in one version chemistry, only that output is reported. eQTLs with a gene-level FDR < 0.05 were considered statistically significant, and a permutation-based strategy (n = 10) we had described before was used to control this FDR[2]. Using the same parameters described above, but without eQTLGen SNP-gene pair filtering, we also performed a genome-wide cis-eQTL discovery analysis.

Next, we performed re-QTL mapping, confining ourselves to the total gene set of FDR < 0.05 significant eQTLs across all cell types and conditions. For this, we calculated the log ratio of the averaged expression of the unstimulated condition and the stimulated condition per sample, cell type, and chemistry, and then applied the same mapping strategy we used to identify regular eQTLs.

To determine whether eQTLs and re-QTLs were genetically inflated, eQTLgen lead-eSNPs were matched to the top GWAS SNP per locus for each of the following immune-mediated disease GWAS studies: celiac disease[69], type 1 diabetes[46], multiple sclerosis[70], inflammatory bowel disease[68], candidemia susceptibility[72] and rheumatoid arthritis[45]. For this, the LD between eSNPs and GWAS SNPs was calculated from genotypes of the 503 European individuals in the 1000 g phase3 reference panel at R2 > 0.8 using Plink 1.9-beta6[87]. Lambda inflation was calculated using all GWAS p-values matched to the eQTL or re-QTL SNPs. To determine whether there is a difference in genomic inflation for those SNPs whose eQTL effect changes upon stimulation (re-QTLs), we compared the genomic inflation of the re-QTL SNPs with the non-re-QTL overlapping eQTL SNPs that were tested in both the unstimulated and relevant stimulation condition and significant in either. Using the different conditions and GWASes, specifically for the monocytes, the distributions of lambda values for the re-QTL and non-re-QTL sets were compared using a two-sided Wilcoxon Rank Sum Test. This statistical testing was solely performed in monocytes, as this was the cell type with a strong pathogen response and the largest set of identified re-QTL SNPs, expecting the largest effects on genomic inflation and allowing for the most robust genomic inflation analysis.

### Co-expression QTLs: mapping, pathway enrichment, TFBS, and GWAS overlap.
Co-expression QTL mapping was performed in the monocytes on a subset of SNPs and genes, selected based on their being: (1) DE and (2) a re-QTL in at least one of the stimulation-timepoint combinations; (3) expressed in at least 50% of the individuals (for each 10x chemistry tested). This selection resulted in 49 SNP-gene combinations for which we calculated the Spearman correlation with every other gene per individual and per stimulation-timepoint condition. A weighted linear model was used in which the genotype predicts the strength of the correlation between the two genes, using the square root of the number of cells as a weight. Analysis was performed separately for the different 10x chemistries, after which betas and standard errors were meta-analyzed. When an analysis could only be performed in one version chemistry, only that output is reported. The statistical significance threshold was then determined using a permutation-based (n = 100) FDR approach. The most significant co-expression QTL p-value per stimulation-timepoint condition was then compared with the one coming from re-running the same permutations after randomly shuffling the genotype identifiers. This allowed us to calculate an eQTL gene-level FDR[2]. An FDR < 0.05 was considered statistically significant. Separate thresholds were determined for each re-QTL SNP-gene combination and each stimulation-timepoint condition.

Pathway analysis was performed on the co-expression QTL genes associated with the selected eQTL gene per stimulation-timepoint combination using Toppfun with similar settings to those described in the 'DE and pathway analysis' section. Significant pathways (BH-corrected p-value < 0.05) were then ranked by p-value. The rankings of the pathways for each condition were then clustered using hierarchical clustering using the complete linkage method.

Transcription factor motif enrichment analysis was performed on the 3 h stimulation outer join CLEC12A co-expressed gene set split by having either a more positive or more negative correlation with the minor versus major allele. For this, we took information from the TRANSFAC database release 2020.2 v2[54] and used g:Profiler (version e102_eg49_p15_7a9b4d6)[88] with the g:SCS multiple testing correction method, applying a significance threshold of 0.05. Additionally, the CLEC12A co-expression QTL SNP rs12230244 and its accompanying (near-) perfect LD SNPs were overlapped with putative TFBSs, as defined by SNP2TFBS[55].

Overlap of co-expression QTL SNPs (or SNPs within a 1 Mb window with LD > 0.8) with disease-associated GWAS SNPs was determined by searching the GWAS catalog (https://www.ebi.ac.uk/gwas/) and an additional set of immune-related GWAS studies (celiac disease[69], type 1 diabetes[46], multiple sclerosis[70], inflammatory bowel disease[68], candidemia susceptibility[72] and rheumatoid arthritis[45]).

### CLEC12A co-expression QTL validation and replication: SLE PRS interaction analysis, SLE scRNA-seq co-expression QTL analysis.
Using the summary statistics of the SLE GWAS by Bentham et al.[67], we calculated the PRS for SLE in 3553 samples from the BIOS consortium using a custom Java program, GeneticRiskScoreCalculator-v0.1.0c, as described previously[56]. Briefly, to account for LD between variants, our approach included a double clumping strategy where we first clumped variants within a 250 kb window and then within a 10 Mb window using an LD threshold R2 = 0.1. We then calculated the PRS for each individual by summing the products of the number of risk alleles and the GWAS effect size (i.e.,

beta) for each SLE-associated variant. We constructed the PRS using a *p*-value threshold for the SLE GWAS of $p < 5 \times 10^{-8}$. The resulting PRS was scaled between 0 and 2 for compatibility with the eQTL-mapping software. We then determined whether the co-expression between *CLEC12A* and an individual's SLE PRS was modulated by SNP rs12230244. For this, we fitted a generalized linear model with and without the SLE PRS as an interaction term and determined how far the predicted model deviated from the true observation by taking the residuals of the observation. A one-tailed F-test was then used to determine whether the distribution of the squared residuals with the SLE PRS as an interaction term was significantly smaller than without, meaning that the SLE PRS interacts with the *CLEC12A* co-eQTL.

We used an independent cohort of SLE patients and healthy controls (GEO accession number: GSE174188) to replicate our findings of a clear enrichment for IFN-related genes within the co-expressed gene set of the *CLEC12A*-SNP rs12230244 co-expression-QTL[61]. This cohort contained individuals of EUR and EAS descent, including healthy individuals (EAS: 18, EUR: 58) and individuals diagnosed with SLE (EAS: 58, EUR:59) who were not in an active disease state when samples were collected. For all individuals, PBMCs were collected and cryopreserved until further use. The SLE samples were collected through the California Lupus Epidemiological Study (CLUES) cohort. Healthy controls were collected at the UCSF Rheumatology Clinic and through the Immune Variation Consortium (ImmVar) in Boston. All UCSF samples were genotyped using the Affymetrix World LAT Array, while samples collected in Boston were genotyped using the Illumina OmniExpressExome Array. The Michigan Imputation Server was used for imputation with the Haplotype Reference Consortium version 1.1 reference. The samples collected at UCSF and Boston were processed using established protocols[11,27]. ScRNA-seq was performed using 10x Chromium Single Cell 3' V2 chemistry, as described previously[11]. Libraries were sequenced on the HiSeq4000 or NovaSeq6000 at a depth of 6306–29,862 reads/cell. Freemuxlet was used to assign cells to individuals and, together with Scrublet[89], for the identification of doublets. Marker gene expression was used to assign the major cell types. Only the monocytes were taken for this independent discovery analysis. Monocytes with less than 1500 UMIs were removed, as were donors with fewer than 200 cells remaining after applying this cutoff. Co-expression QTL analysis was performed as described in the co-expression QTL mapping paragraph above, but only testing the *CLEC12A*-SNP rs12230244 co-expression-QTL and doing this analysis separately in each cohort, ancestry, and disease state (SLE versus healthy). A meta-analysis of the cohorts and ancestries was then performed, and pathway analysis using the REACTOME database was conducted to determine whether the IFN pathway was differently enriched in SLE compared to the healthy controls.

**Reporting summary**. Further information on research design is available in the Nature Research Reporting Summary linked to this article.

## Data availability

The number of individuals and cells included in each analysis can be found in the Source Data file. Raw gene expression counts, eQTL, and co-expression QTL summary statistics can be found under "Supplementary Data" at the website accompanying this paper (https://eqtlgen.org/sc/datasets/1m-scbloodnl.html). Processed (de-anonymized) scRNA-seq data, including a text file that links each cell barcode to its respective individual, has been deposited at the European Genome-Phenome Archive (EGA), which is hosted by the EBI and the CRG, under accession number EGAS00001005376. Gene expression and genotype data can be obtained and requested by filling in a short web form at https://eqtlgen.org/sc/datasets/1m-scbloodnl.html. This form is subsequently reviewed by a single Data Access Committee, who will be able to approve access to both the raw gene expression and genotype data within 5 working days (during the holiday season there might be a slight delay). Once the proposed research is approved, access to the relevant gene expression or genotyped data will be free of charge. Access to the genotype and gene expression data is facilitated via the HPC cluster of the UMCG and the EGA, respectively. Access to this data is restricted to comply with the European Union General Data Protection Regulation for protection of privacy-sensitive data. Sample metadata (age, gender) is presented in Supplementary Data 1. The REACTOME and TRANSFAC release 2020.2 v2[54] database can be accessed through https://reactome.org/ and https://biit.cs.ut.ee/gprofiler/gost, respectively.

## Code availability

The original code for Seurat v3.1[81] (https://github.com/satijalab/seurat), Eagle v2.322 (https://github.com/poroloh/Eagle), Demuxlet[11] 85dca0a4d648d18e6b240a2298672394fe10c6e6 (March 25, 2019) (https://github.com/statgen/demuxlet), Souporcell v1.0[29] (https://github.com/wheaton5/souporcell), Freemuxlet v1 as part of the Popscle suite of statistical genetics tools (https://github.com/statgen/popscle), Scrublet[89] v0.2 (https://github.com/swolock/scrublet), the GeneticRiskScoreCalculator v0.1.0c[56] (https://github.com/molgenis/systemsgenetics/tree/master/GeneticRiskScoreCalculator), and our in-house eQTL pipeline[2] v1.4.0 (https://github.com/molgenis/systemsgenetics/tree/master/eqtl-mapping-pipeline) can be found at GitHub. All custom-written code is made available via GitHub (https://github.com/molgenis/1M-cells).

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

## Acknowledgements

We are very grateful to all the volunteers who participated in this study and would like to thank K. McIntyre for proofreading the manuscript. This work was carried out on the computer cluster of the Genomics Coordination Center, hosted at the University of

Groningen Center for Information Technology. M.G.G. was supported by the National Institutes of Health under Ruth L. Kirschstein National Research Service Award F31HG011007. C.J.Y. is further supported by the NIH grants R01AR071522, R01AI136972, U01HG012192, and the Chan Zuckerberg Initiative, and is an investigator at the Chan Zuckerberg Biohub and is a member of the Parker Institute for Cancer Immunotherapy (PICI). L.F. and M.W. are supported by grants from the Dutch Research Council (ZonMW-VIDI 917.14.374 and ZonMW-VICI to L.F., NWO-VENI 192.029 to M.W.), and by an ERC Starting Grant, grant agreement 637640 (ImmRisk) and through a Senior Investigator Grant from the Oncode Institute. The Biobank-Based Integrative Omics Studies (BIOS) Consortium is funded by BBMRI-NL, a research infrastructure financed by the Dutch government (NWO 184.021.007). The images in Fig. 4f are created using Servier Medical Art, which we are thankful to for providing free online images.

## Author contributions

M.W. collected blood samples and generated the scRNA-seq data. R.O., D.V., and H.B. performed bioinformatics and statistical analyses. R.O., D.V., H.B., and M.W. generated figures. H.B. built the website accompanying the manuscript. M.G.G. and C.J.Y. performed and provided SLE scRNA-seq data for co-expression QTL replication analysis. M.V. and H.W. provided critical input for the statistical analyses. The BIOS consortium provided samples to conduct the SLE PRS co-expression QTL analysis. D.V., M.W., and L.F. designed the study. R.O., D.V., and M.W. wrote the manuscript and all other authors provided critical feedback. All authors discussed the results and commented on the manuscript.

## Competing interests

C.J.Y. is a Scientific Advisory Board member for and hold equity in Related Sciences and ImmunAI, a consultant for and hold equity in Maze Therapeutics, and a consultant for TReX Bio. C.J.Y. has received research support from Chan Zuckerberg Initiative, Chan Zuckerberg Biohub, and Genentech. The remaining authors declare no competing interests.

## Additional information

## single-cell eQTLGen consortium

Roy Oelen [1,2,11], Harm Brugge[1,2,11], M. Grace Gordon [3,4,5,6], Chun J. Ye [4,6,7,8,9,10], Lude Franke[1,2,12✉] & Monique G. P. van der Wijst [1,2,12✉]

## BIOS Consortium

Lude Franke[1,2,12✉]

A full list of members and their affiliations appears in the Supplementary Information.

