## [Peer Review File · Nature Communications]

Single-cell RNA-sequencing of peripheral blood mononuclear cells reveals widespread, context-specific gene expression regulation upon pathogenic exposureREVIEWER COMMENTS

Reviewer #1 (Remarks to the Author):

The current study describes an in-depth single-cell RNA-seq based study of immune cells and their responses to exposure to several important pathogenic bacteria.

The overall concept is interesting and the findings have the potential to be very valuable. However, I have some substantial concerns with the scRNA-seq analytics and I feel these have to be addressed before there is any value in looking in detail at the other findings of the manuscript.

The study uses the 10x platform for the scRNA-seq work. This is a broadly used technology and I have no issue with the platform from that perspective. However, the one drawback of this method is that older chemistries can give relatively low coverage per cell. This is illustrated in Figure S1B. Using the V2 chemistry, the authors appear to achieve a maximum of about 80,000 reads per cell, but the vast majority of cells appear to have much lower coverage (<10,000 reads per cell). It's a little hard to see precisely what proportion of cells achieve what level of coverage from the plot and I wonder if the authors might also add a frequency histogram and a supplementary table to assist with being able to better communicate this to the reader.

This low coverage presents issues with gene dropout (genes going unmeasured due to low read coverage). As shown in Figure S1B again, for the v2 chemistry, when gene detects to read coverage curve begins to flatten out after about 20,000 reads per cell. However, the authors mention only filtering out cells with fewer than 200 gene detects. It's not easy to eye-ball this from the figure, but that would appear to allow the vast majority of cells to be kept in the analysis, even when they are likely to have >90% gene dropout.

The V3 chemistry gives higher read coverage. I can't read the exact number of the figure as the version of the figure I have access to isn't high enough resolution, but it's clearly much higher. Consistent with this, the authors have cells with up to ~10,000 detected genes. Still, based on the heat map legend provided, it appears the vast majority of cells have < ~1000 detected genes, again pointing to issues with gene dropout. I think this high rate of gene dropout may have impacted on the DE analyses. It's not explicitly stated, but given the many blank cells in Table S2 and the extremely low bonferroni scores for a number of genes (the vast major have a value of 0), I suspect missing data due to gene dropouts have had a significant impact.

It's not reasonable to expect the authors to sequence all cells in their dataset to a level needed to approach gene saturation. This would be exceedingly expensive and ridiculous. However, I do think they need to provide more information about the number of missing genes per cell and the rate of gene loss/dropout by read coverage. This should be used as a guide for setting the minimum coverage threshold for the cells to be kept and used in the DE analyses. The authors have a very large number of cells in their dataset (>920,000 that pass their current QC filters). Personally, I'd rather see a dataset of 100,000 deeply sequenced cells over one including a larger number of less well-covered ones. From that perspective, if the authors discarded substantially more cells in the QC stage, I don't think it would detract at all from the study, provided it didn't mean losing the ability to resolve and differentiate each cell type.

In addition, there are many studies describing the need to use data imputation to account for gene dropouts, particularly for lower coverage datasets/cells. I don't see anything in the methods suggesting this has been used here. This should be included in conjunction with the minimum coverage thresholds the authors' use. A more permissive coverage cut-off would likely need a more stringent approach the imputing missing data and vice versa.

I think these issues can be overcome through further informatic analysis and if the authors' findings hold up after having addressed this, I do think the study is well worth further consideration for Nature Communications. Without this however, I don't see that the findings can be viewed with enough confidence. In my opinion, this issue needs to be addressed before a detailed review of the rest of the paper is possible.

I'm sorry I can't be more supportive at this time. I do hope to see a revised version of the study, but in my opinion it needs major revision first.

Reviewer #2 (Remarks to the Author):

The manuscript by Oelen et al. reports that PBMCs being stimulated by *C. albicans*, *M. tuberculosis* or *P. aeruginosa* showed the potential regulations of gene expression, eQTL effects, and context-specific QTL. It is an interesting and meaningful study, which conducted a big scale of scRNA-seq using PBMCs stimulated with different pathogens. Besides, the authors identified the potential regulation of the IFN pathway and IRF1 for CLEC12A, as well as analyzed eQTLs and QTLs in different infections, which may provide a novel strategy for finding novel research targets. Some concerns are needed to be addressed.

Major concerns:

1. Please provide the detailed biomarkers for identifying these cell clusters in heatmap or dot plots (Line 116).
2. Line 127: The authors claimed that two batches were sequenced and mixed cells in sequence analyses. How did the authors correct the batch effects? Are the functions of the Seurat package (Line 593) enough to decrease the batch effects?
3. In this manuscript, the authors focused on eQTLs and QTLs of pathogen infections. It would be better to correlate these findings with clinical phenotypes in all these individuals, especially their different genetic susceptibilities to these pathogens.
4. The results revealed cells with the largest DE showed lower eQTLs. Could it be explained by some specific transcript factors up-regulated in these cells? An analysis of transcription factor viability among different cell clusters would improve this study (Line 252).
5. The authors proposed that the IFN pathway is involved in the regulation of CLEC12A through SNP rs12230244, with a series of analyses. More functional studies are needed to support this claim, showing how high viability of IFN signaling will up-regulate CLEC12A.
6. Last, the authors should highlight the key clinical values of these findings.

Minor concerns:

1. It would be better to use unique DE genes rather than all DE genes for each cell type in the pathways analyses (Line 190).
2. Character overlaps in some supplementary figures, for example, Figure S5.

Reviewer #3 (Remarks to the Author):

Oelen et al. present scRNA-seq analysis for PBMCs from a cohort of 120 individuals, and identify the presence of both cell-type-specific eQTLs and response eQTLs. This is an important work as so far there have been few single-cell population genetic studies aiming to find cell-type-specific eQTLs, and those that have been published profiled smaller populations. The analysis and results presented are well executed, and it was particularly interesting to see that response-specific eQTLs were often shared across the different pathogens, and decreased over time. I have a few comments which I hope can further improve the current manuscript.

A major goal of the study was to identify the cell types in which eQTLs are active. My main question is whether single-cell population genetic studies are required to determine the cell types implicated in eQTLs discovered from bulk-cell assays, or whether the cell types involved can be accurately inferred from other types of data. For example, for eQTL genes, you could find the list of cell types expressing that gene using single-cell RNA-seq data from a single individual. You could also look at scATAC-seq data from a single individual and examine which cell types had open ATAC peaks near or overlapping the variant position. In generating the single-cell population genetic dataset here, the authors are uniquely able to address this question, which I feel is valuable to the field: should we be pursuing single-cell population genetic experiments, or is it more effective to do targeted single-cell experiments on a small number of individuals and "deconvolve" signals found using bulk assays on much larger populations.

The analysis presented here has focused on six major PBMC cell types, but there are much more

than this present in PBMCs (naive, memory CD4/CD8 T cells, pro/pre B cells, CD14/CD16 monocytes, etc.). Given the large sample size (across all donors), I am unsure why the authors have restricted their analysis to such broad cell type classifications. The major advantage of a single-cell assay is the ability to identify all of these different cell types in the dataset.

Minor comments:

Figure 2e is unclear (especially the black and white heat map at the top). Perhaps splitting genes by group and constructing separate heat maps would be more clear.

Figure 4b is unclear. I think the top plots are showing expression of CLEC12A vs PML, with each point being a cell, but I'm unsure why they're shown as lines originating at zero and why the data appear to be discrete on the x-axis?

Several details in the single-cell RNA-seq analysis can be explained better. For example (line 593), how were integration anchors found? Wherever possible, the name of the function used and the parameters need to be reported.

How were cell types annotated? What marker genes were used and what was the process of assigning cells to cell types based on these marker genes?

Where is the GeneticRiskScoreCalculator program available?

Response to reviewer comments

We would like to thank the reviewers for their highly constructive comments and suggestions, which have enabled us to improve our manuscript substantially. Below we will provide a response (plain text) to each reviewer comment (bold) with specific adjustments to the text being italicized.

Reviewer #1 (Remarks to the Author):

The current study describes an in-depth single-cell RNA-seq based study of immune cells and their responses to exposure to several important pathogenic bacteria.

The overall concept is interesting and the findings have the potential to be very valuable. However, I have some substantial concerns with the scRNA-seq analytics and I feel these have to be addressed before there is any value in looking in detail at the other findings of the manuscript.

The study uses the 10x platform for the scRNA-seq work. This is a broadly used technology and I have no issue with the platform from that perspective. However, the one drawback of this method is that older chemistries can give relatively low coverage per cell. This is illustrated in Figure S1A. Using the V2 chemistry, the authors appear to achieve a maximum of about 80,000 reads per cell, but the vast majority of cells appear to have much lower coverage (<10,000 reads per cell). It's a little hard to see precisely what proportion of cells achieve what level of coverage from the plot and I wonder if the authors might also add a frequency histogram and a supplementary table to assist with being able to better communicate this to the reader.

First of all we have to apologize for the ambiguous labeling of the x-axis in Fig. S1A: instead of 'number of reads', the x-axis should read 'number of unique reads (or number of UMIs)', which we have now relabeled accordingly. This also affects the interpretation of the figure, as the number of UMIs found in individual cells is very much related to the cell type (and size). For example, T cells are relatively small cells and therefore express a relatively small number of genes and UMIs. While on the other hand, monocytes or plasmablast are relatively large and express a much larger number of genes and UMIs. This has previously been shown in literature (see for example Fig. 8D of Yamawaki et al. 2021 - BMC Genomics: <https://bmcbgenomics.biomedcentral.com/articles/10.1186/s12864-020-07358-4>). In other words, the biological heterogeneity of a PBMC sample explains for a large part the huge spread in number of UMIs (and number of Genes), and therefore excluding cells based on the number of UMIs would mean removing biological heterogeneity/excluding particular cell types. We have now visualized this in a new Fig. S1D that shows boxplots of the nUMIs split by cell type. Additionally, to further assist the reader interpreting the data we have added a new Fig. S1C showing a frequency histogram of the number of detected UMIs. Please note that the combination of QC cut-offs used in this study (high mtDNA-encoded gene percentage, low nGenes, doublets) is chosen in such a way that low quality cells are excluded without removing the biological heterogeneity.

This low coverage presents issues with gene dropout (genes going unmeasured due to low read coverage). As shown in Figure S1A again, for the v2 chemistry, when gene detects to read coverage curve begins to flatten out after about 20,000 reads per cell. However, the authors mention only filtering out cells with fewer than 200 gene detects. It's not easy to eye-ball this from the figure, but that would appear to allow the vast majority of cells to be kept in the analysis, even when they are likely to have >90% gene dropout.

In the previous version of Fig. S1A the cells with fewer than 200 genes expressed were already removed. To visualize how many cells will be removed by this QC step, we have now updated this figure - Fig. S1B - by including all cells before filtering and by adding a red line showing the 200

nGene cut-off. To provide additional insights, we now also include Table S3 describing the number of cells that are lost during each QC step, and split by version chemistry.

The reviewer is correct that the average percentage of zeros in the count matrix exceeds 90%: 95.47% of the count matrix were zeros for the v2, and 91.56% for the v3 chemistry data. This is a normal observation for 10X Genomics scRNA-seq data from PBMCs, which are cells that by themselves are on the lower end of RNA content (PBMCs are so-called 'RNA-poor cells'). For example, the publicly available 3k PBMC v1 10X dataset (from 10X Genomics themselves) contains 97.41% zeros, and our previously published PBMC v2 10x scRNA-seq dataset contained 94.1% zeros [Van der Wijst 2018 - Nature Genetics]. In this study we used the sequencing saturation parameter from the CellRanger output file as a guidance to define which samples had to be sequenced at higher depth after the first round of sequencing to reach a good balance between the sequencing saturation and the associated cost. We aimed for a sequencing saturation of at least 75% for v2 (~37k reads/cell on average), and at least 60% for v3 chemistry libraries (~65k reads/cell on average). We have now added a new Table S2 in which we describe the QC parameters extracted from the Cellranger output, including the number of cells, average number of reads, genes and UMIs per cell, the fraction of reads within a cell and the sequencing saturation.

The V3 chemistry gives higher read coverage. I can't read the exact number of the figure as the version of the figure I have access to isn't high enough resolution, but it's clearly much higher. Consistent with this, the authors have cells with up to ~10,000 detected genes. Still, based on the heat map legend provided, it appears the vast majority of cells have < ~1000 detected genes, again pointing to issues with gene dropout. I think this high rate of gene dropout may have impacted on the DE analyses. It's not explicitly stated, but given the many blank cells in Table S2 and the extremely low bonferroni scores for a number of genes (the vast major have a value of 0), I suspect missing data due to gene dropouts have had a significant impact.

The blank cells/missing values in the original Table S2 (now Table S5) mean that a gene is not tested or not meta-analyzed. This is the case when a gene was not expressed in at least 10% of the cells that were used for a specific DE analysis. The very low P-values are the result of the large number of cells that were used for the DE analysis, which as a consequence provided large statistical power for the DE analysis: for the smallest cell type-condition combination (i.e. the DCs in the 24h MTB v3 condition) about 750 cells were used for the DE analysis, going up to over 30,000 cells for the largest cell type-condition combination (i.e. the CD4T cells in the UT v2 condition). Based on a recent study in which simulated scRNA-seq data was used to define the overall detection power for DE analysis using a certain number of input cells, we expect to have an overall detection power of ~0.15 in our smallest cell type-condition combinations, but this goes up to ~0.75 in our largest cell type-condition combinations (Fig. 3 of Schmid et al. 2021 <https://www.nature.com/articles/s41467-021-26779-7>).

The reviewer is right that dropouts may impact our power to detect DE genes, and that the level of dropouts can differ per experimental batch. However, to prevent that such batch effects cause false positive results, we designed the experiment in such a way that both untreated and stimulated cells were processed together in batches of 8 samples each. As such, if dropouts would have impacted the DE analysis, then this would not be related to any of the stimulation conditions. Therefore, it is unlikely that the DE results are the consequence of dropouts.

It's not reasonable to expect the authors to sequence all cells in their dataset to a level needed to approach gene saturation. This would be exceedingly expensive and ridiculous. However, I do think they need to provide more information about the number of missing genes per cell and the rate of gene loss/dropout by read coverage. This should be used as a guide for setting the minimum coverage threshold for the cells to be kept and used in the DE analyses. The authors have a very large number of cells in their dataset (>920,000 that pass their current QC filters). Personally, I'd rather see a dataset of 100,000 deeply sequenced cells over one including a larger number of less well-covered ones. From that perspective, if the authors discarded substantially more cells in the

QC stage, I don't think it would detract at all from the study, provided it didn't mean losing the ability to resolve and differentiate each cell type.

As mentioned before, the number of UMIs or genes detected per cell is actually closely related to the underlying cell type/biological heterogeneity within PBMCs (as now shown in Fig. S1D). On top of that, as shown in Figure 3 of the recent study from Schmid et al. 2021 (<https://www.nature.com/articles/s41467-021-26779-7>), decreasing the number of cells used as input for the DE analysis will lower the overall detection power. On the other hand, several benchmarking studies have shown that our chosen DE analysis method (MAST), does not identify many false positives [Soneson et al. 2018 - Nature Methods <https://www.nature.com/articles/nmeth.4612>; Wang et al. 2019 - BMC bioinformatics <https://doi.org/10.1186/s12859-019-2599-6>] and does not enrich for genes with a large fraction of zeros [Soneson et al. 2018 - Nature Methods <https://www.nature.com/articles/nmeth.4612>]. For all these reasons together, we believe that such stringent cut-offs for the number of UMIs detected per cell would not improve our DE analysis.

Nevertheless, we do agree that it is useful for the reader to gain more insights in the general quality of the dataset. Therefore, we have now included a new Table S2 that describes the general QC parameters from the Cellranger output, including the number of cells, average number of reads, genes and UMIs per cell, the fraction of reads within a cell and the sequencing saturation. It is also important to note that we used the sequencing saturation parameter from the CellRanger output file as a guidance to define which samples had to be sequenced at a higher depth after the first round of sequencing to reach a good balance between the sequencing saturation and the associated cost. We aimed for a sequencing saturation of at least 75% for v2 (~37k reads/cell on average), and at least 60% for v3 chemistry libraries (~65k reads/cell on average).

In addition, there are many studies describing the need to use data imputation to account for gene dropouts, particularly for lower coverage datasets/cells. I don't see anything in the methods suggesting this has been used here. This should be included in conjunction with the minimum coverage thresholds the authors' use. A more permissive coverage cut-off would likely need a more stringent approach the imputing missing data and vice versa.

We have not used gene expression imputation in this study, as several studies provide reasons to believe it is not benefitting our analyses. Firstly, a DE benchmarking study showed that the chosen DE method (MAST) does not bias for genes with a higher fraction of zeros [Fig. 2: Soneson et al. 2018 - Nature Methods <https://www.nature.com/articles/nmeth.4612>]. Secondly, another recent benchmarking study on imputation methods showed that while some imputation methods improve detecting DE genes, they also can introduce false positive signals [Hou et al., 2020 - Genome Biology - <https://genomebiology.biomedcentral.com/articles/10.1186/s13059-020-02132-x>]. Moreover, most imputation methods amplify large effect sizes compared to no imputation. However, if the original expression difference is small, then most imputation methods may remove such small differences and hence do not show a clear advantage over not imputing. Therefore, taking this all together, with the large sample size of our study, we do not require imputation to detect the large DE effects and it is even better to not impute for the small DE effects.

I think these issues can be overcome through further informatic analysis and if the authors' findings hold up after having addressed this, I do think the study is well worth further consideration for Nature Communications. Without this however, I don't see that the findings can be viewed with enough confidence. In my opinion, this issue needs to be addressed before a detailed review of the rest of the paper is possible.

I'm sorry I can't be more supportive at this time. I do hope to see a revised version of the study, but in my opinion it needs major revision first.

Reviewer #2 (Remarks to the Author):

The manuscript by Oelen et al. reports that PBMCs being stimulated by *C. albicans*, *M. tuberculosis* or *P. aeruginosa* showed the potential regulations of gene expression, eQTL effects, and context-specific QTL. It is an interesting and meaningful study, which conducted a big scale of scRNA-seq using PBMCs stimulated with different pathogens. Besides, the authors identified the potential regulation of the IFN pathway and IRF1 for CLEC12A, as well as analyzed eQTLs and QTLs in different infections, which may provide a novel strategy for finding novel research targets. Some concerns are needed to be addressed.

Major concerns:

1. Please provide the detailed biomarkers for identifying these cell clusters in heatmap or dot plots (Line 116).

We have now included Table S4 that describes the marker genes that were used to define the major cell types that were identified in this study.

2. Line 127: The authors claimed that two batches were sequenced and mixed cells in sequence analyses. How did the authors correct the batch effects? Are the functions of the Seurat package (Line 593) enough to decrease the batch effects?

At different levels in our dataset batch effects could have been introduced. The three most important ones, going from the largest to smallest batch effect, are: 1. 10x version chemistry differences (v2 versus v3 chemistry); 2. experimental date of the library prep; 3. lane within a 10x chip (per 10x chip we loaded two lanes in one run). The latter effect was mostly described already by experimental date, so we did not further correct for the experimental lane itself in downstream processing.

The first batch effect induced by version chemistry could not be corrected for using the functions of the Seurat package or any other batch correction tool. Therefore, instead of correcting for it, we ran all downstream analyses for the v2 and v3 chemistry separately, and then meta-analyzed this data.

For the DE results (p30, line 657-660): *"We used MetaVolcanoR [Cesar Prada 2021] to perform a meta-analysis for each cell type, taking the results of the v2 and v3 chemistries as inputs for Fishers Combined Probability Test [Fisher RA 1932]. Significance was determined by taking a Bonferroni-corrected p-value of <0.05 within the meta-analysis."*

For the eQTL results (p31, line 689-693): *"This was followed by a sample-weighted meta-analysis [Whitlock 2005] over the v2 and v3 chemistry outputs per cell type and stimulation-timepoint combination. eQTLs with a gene-level FDR < 0.05 were considered statistically significant, and a permutation-based strategy (n = 10) we had described before was used to control this FDR [Westra et al. 2013]."*

The batch effect induced by the experimental date was corrected for using normalization strategies: Seurat's SCTransform function [Hafemeister and Satija 2019] or LogNormalize function (scale.factor = 10,000) was used for correction of counts used for eQTL or DE analysis, respectively.

We now describe this in more detail in the materials and methods scRNA-seq alignment, preprocessing and QC, (p29, line 639-644): *"For annotating the cell types, we first log-normalized the count matrices for each of the seven timepoint-stimulation conditions and two chemistries separately using Seurat's LogNormalize function (scale.factor = 10,000) [Hafemeister and Satija 2019] The log-normalized count matrices of the unstimulated data were then integrated separately for each of the*

three pathogen stimulations. For this, we used the first 30 principal components (PCs) to identify integration anchors using Seurat's `FindIntegrationAnchors` function. These anchors were then used for integration using Seurat's `IntegrateData` function [Hafemeister and Satija 2019]".

3. In this manuscript, the authors focused on eQTLs and QTLs of pathogen infections. It would be better to correlate these findings with clinical phenotypes in all these individuals, especially their different genetic susceptibilities to these pathogens.

The hypothesis of our study is that genetic effects can interact with environmental factors to make individuals more susceptible to disease. Looking at response-QTLs, we learn for which genes a combination of genetics and environment (of a pathogen infection) can affect the gene expression level. And subsequent enrichment analysis with GWASes shows that these response-QTLs are more often associated with disease. The reviewer is suggesting another way to look at this: by correlating gene expression to clinical phenotypes. However, in this study we cannot correlate expression patterns to clinical phenotypes, as the individuals within the used cohort (Lifelines DEEP) are from the general population (so in general they are healthy). Moreover, this cohort is not linked to any electronic health record information, nor is the current disease status of these individuals known.

What could be done in this cohort, is that polygenic risk scores (PRSs) can be calculated for clinical phenotypes related to the pathogen stimulations (i.e. candidemia, tuberculosis, etc) or any other trait. PRSs are built from taking the sum of all known (protective or detrimental) SNPs for a particular trait. These PRSs can then be linked to gene expression levels (eQTS), to help prioritize potential trait-relevant genes. However, a severe limitation of PRSs is that they are as good as their input information. And so as GWASes for these types of diseases are still highly underpowered [178 cases Candidemia: Jaeger et al. 2019; 1,768 cases UK Biobank TB], the calculated PRS would be as well. Moreover, even if the relevant GWAS would be sufficiently powered, it is still unlikely that we have sufficient statistical power to detect (many) such eQTS effects in our cohort of 120 individuals: eQTS effects are usually rather small and require more power to detect than for a regular *cis*- or *trans*-eQTL effect (see for example a recent study by the eQTLGen consortium performed in whole blood samples of 31,684 individuals, in which they identified *cis*-eQTLs for 16,987 genes, *trans*-eQTLs for 6,298 genes and eQTS effects for 2,568 genes) [Vösa et al. 2021].

Nevertheless, we do like the suggestion and believe that this type of analysis can be very interesting to conduct when scRNA-seq sample sizes increase further. A few years ago we have set up the sc-eQTLGen consortium (van der Wijst et al. 2020) with the goal to make a standardized pipeline to process and eventually combine all these scRNA-seq population-based datasets with the aim to pinpoint the cellular contexts in which disease-causing genetic variants affect gene expression. Within this consortium, we currently have scRNA-seq data available for over 2,000 individuals. We believe that such efforts open up the opportunity to conduct sc-eQTS analyses in the future, and may further uncover how particular contexts are associated with disease.

We now discuss this opportunity in the Discussion and Conclusion section of the manuscript (p27, line 589-592): "*Moreover, instead of linking individual genetic variants, linking of polygenic risk scores to cell-type-specific gene expression (i.e. eQTS analysis [Vösa et al. 2021]) may provide a more disease-focused insight into how the combination of disease-associated variants together contribute to changes in gene expression levels.*"

4. The results revealed cells with the largest DE showed lower eQTLs. Could it be explained by some specific transcript factors up-regulated in these cells? An analysis of transcription factor viability among different cell clusters would improve this study (Line 252).

In this study, we used co-expression QTL analysis to dissect the context in which gene expression is being regulated. Our approach is to do this without prior knowledge with regards to transcription factors, as we believe that a targeted search (i.e. transcription factor-eQTL approach as described in a

recent preprint by Flynn et al. [Flynn et al. 2021, <https://www.biorxiv.org/content/10.1101/2021.07.20.453075v2>]) would introduce biased assumptions. The reason for this is that most ‘knowledge’ is previously collected in bulk samples, but that information at the bulk level does not have to be true at the single-cell level or in a specific cellular or environmental context. Moreover, in contrast to the preprint by Flynn et al. in which unstimulated bulk samples were used to identify transcription factor-eQTLs, our scRNA-seq dataset contains PBMCs exposed to different pathogens. At the resolution our scRNA-seq data is generated, existing knowledge is rather limited, and as such, cannot extensively be used for a transcription factor-eQTL like approach.

Another worry we have regarding the suggested approach is that we expect that in the context of pathogen stimulations, genes that are regulated by transcription factors cannot be uncovered by looking at the mRNA expression levels of the transcription factors. The reason being that many of those pathogen-induced transcription factors have to respond quickly to pathogen exposure, and therefore are already present in the cytoplasm [Liu et al. 2018 - Cell communication and Signaling - <https://biosignaling.biomedcentral.com/articles/10.1186/s12964-018-0224-3>]. It is mainly the translocation from cytoplasm to nucleus that regulates the active component of these transcription factors. Therefore, instead of looking at mRNA expression levels, ideally nuclear protein levels should be used. Several recent preprints describe methodology that would allow measuring scRNA-seq data and the nuclear transcription factor protein levels from the same single cells (e.g. inCITE-seq [Chung et al. 2021, BiorXiv - <https://www.biorxiv.org/content/10.1101/2021.01.18.427139v1>] and NEAT-seq [<https://www.biorxiv.org/content/10.1101/2021.07.29.454078v1.full.pdf>]). We believe that data generated with such methodology would be most suitable for applying more biased transcription factor-eQTL-like approaches at the single-cell level and in the context of pathogen stimulations.

We now discuss the above in the Discussion section (p26-27, line 578-582): “*We foresee that newly developed methodology, such as inCITE-seq [<https://www.biorxiv.org/content/10.1101/2021.01.18.427139v1>] and NEAT-seq [<https://www.biorxiv.org/content/10.1101/2021.07.29.454078v1.full.pdf>], combining measurements of multiple omics layers from the same cell, including RNA and nuclear protein levels (which allows measuring active transcription factors levels), will further enhance the interpretability of the identified co-expression QTLs in the future.*”

5. The authors proposed that the IFN pathway is involved in the regulation of CLEC12A through SNP rs12230244, with a series of analyses. More functional studies are needed to support this claim, showing how high viability of IFN signaling will up-regulate CLEC12A.

In this study, we have used three independent datasets to gain evidence that *CLEC12A* is regulated through interferon activity/an interferon regulatory factor (*IRF*): 1. *CLEC12A* co-expression QTL analysis in pathogen-stimulated PBMCs (1M-scBloodNL scRNA-seq dataset), 2. *CLEC12A* co-expression QTL analysis in SLE patients compared to healthy controls (SLE scRNA-seq dataset), 3. *CLEC12A* co-expression QTL analysis against SLE PRS (BIOS consortium whole blood bulk RNA-seq data). Combined with knowledge in literature, this all supported a mechanism through which rs12230244 (or any other SNP in high LD) regulates the binding of IRF in the context of activated IFN signaling, thereby regulating *CLEC12A*.

It is important to note that with the co-expression QTL analysis performed in this study, we aimed to dissect the context in which genes are being regulated. The contexts that we have uncovered contain a combination of having a specific genetic make-up and being exposed to a particular pathogen. In contrast to fine-mapping efforts, our approach is not necessarily designed to pinpoint one specific SNP, although understanding the underlying mechanism of the gene regulation could help to reduce the number of potential candidates. For example, if SNP A, B and C are all in perfect LD with each other, we cannot distinguish which of these SNPs is causally affecting downstream expression of *CLEC12A*. Although understanding the mechanism of action (i.e. IRF binding in the case

of *CLEC12A*), could help reduce the number of candidates (e.g. when IRF only binds to the genomic region in which SNP B and C are located, SNP A can likely be excluded). So, for understanding the mechanism of action or identifying the individuals at risk it does not matter in this case, as individuals at risk will have the risk allele on each of those SNP positions due to the perfect LD between them.

It was not our intention to suggest that we are certain that rs12230244 is modulating the binding efficiency of *IRF*, thereby regulating *CLEC12A* expression. We now make this more explicit in the results and discussion sections where it might have been unclear (p20, line 420): *“Pathway analysis on the CLEC12A co-expression QTL genes revealed stronger enrichment for the IFN pathway in the SLE patients (FDR = 1.965×10⁻⁷) compared to healthy controls (FDR = 1.203×10⁻³), again supporting that this pathway is involved in the regulation of CLEC12A through the locus with the rs12230244 SNP.”*

and (p21, line 465-467): *“This enabled follow-up analyses that gathered solid evidence for the following mechanism of action through which rs12230244 SNP locus affects CLEC12A expression specifically upon 3h pathogen stimulation:”*

and (p26, line 574-576): *“Even though the causal SNP cannot be conclusively determined using co-expression QTL analysis, understanding the underlying mechanism can help to further fine-map the genetic signal.”*

Furthermore, we believe that interpretability of the co-expression QTLs can be further improved in the near future by using methodology that measures multiple single-cell omics layers from the same cell. We now discuss this in the Discussion section (p27-28, line 578-582): *“We foresee that newly developed methodology, such as inCITE-seq [<https://www.biorxiv.org/content/10.1101/2021.01.18.427139v1>] and NEAT-seq [<https://www.biorxiv.org/content/10.1101/2021.07.29.454078v1.full.pdf>], combining measurements of multiple omics layers from the same cell, including RNA and nuclear protein levels (which allows measuring active transcription factors levels), will further enhance the interpretability of the identified co-expression QTLs in the future.”*

6. Last, the authors should highlight the key clinical values of these findings.

The co-expression QTLs identified in our study help to uncover the mechanism that underlies the genetic regulation of particular genes. For example, in the case of *CLEC12A* we uncover that IFN activity, inducing nuclear IRF levels is likely the mechanism that explains the identified co-expression QTLs for rs12230244 on *CLEC12A*. What this implies is that in individuals with the AA allele on this genetic position, conditions that induce IFN activity (e.g. pathogen exposure, the autoimmune disease SLE) will only mildly induce *CLEC12A* expression. In contrast, in individuals with the TT allele on this same genetic position, conditions that induce IFN activity would result in a strong induction of *CLEC12A* expression.

Previously, *CLEC12A* was shown to act as early adaptor molecule for antibacterial autophagy, and in mice, complete knockout of *Clec12a* resulted in higher susceptibility to *Salmonella* infection [Begun et al. 2012, Cell Rep. [10.1016/j.celrep.2015.05.045](https://doi.org/10.1016/j.celrep.2015.05.045)]. Additionally, *CLEC12A* is known to contribute to the pathogenesis of rheumatoid arthritis. For example, upon collagen-induced arthritis, *CLEC12A* knockdown mice show increased joint inflammation [Redelinghuys et al. 2015] and in monocytes of early rheumatoid arthritis patients reduced expression of *CLEC12A* correlated with more severe disease 6 months later [Vaillancourt et al. 2021]. Together, this suggests that individuals with the AA allele on rs12230244 may be at increased risk of bacterial infection and of developing joint inflammation, acting through reduced induction of *CLEC12A* expression when exposed to pathogens or other factors inducing IFN signaling.

We now discuss this clinical relevance in the Results section (p21-22, line 486-495): *“Moreover, looking at the function of the affected genes, we also expect immunological consequences of the identified co-expression QTLs. For example, previously CLEC12A was shown to act as an early adaptor molecule for antibacterial autophagy, and in mice, complete knockout of Clec12a resulted in higher susceptibility to Salmonella infection[Begun et al. 2012]. Additionally, CLEC12A is known to contribute to the pathogenesis of rheumatoid arthritis. For example, upon collagen-induced arthritis, CLEC12A knockdown mice show increased joint inflammation[Redelinghuys et al. 2015] and in monocytes of early rheumatoid arthritis patients reduced expression of CLEC12A correlated with more severe disease 6 months later[Vaillancourt et al. 2021]. Together, this suggests that individuals with the AA allele on rs12230244 may be at increased risk of bacterial infection and of developing joint inflammation, acting through reduced induction of CLEC12A expression when exposed to pathogens or other factors inducing IFN signaling.”*

Minor concerns:

1. It would be better to use unique DE genes rather than all DE genes for each cell type in the pathways analyses (Line 190).

We disagree with the reviewer on this point. There are several biological and technical reasons to not follow this suggestion. First, to confirm that the experiments have succeeded, one could overlay the DE and pathway results with previous literature. As previous literature usually comes from bulk whole blood/PBMC stimulation experiments, you would require all DE genes as input to expect similar output (see Fig. 2D and Fig. S2). Moreover, based on our results, there is quite some sharing of immune genes across the cell types (see Fig. 2B) but this does not mean the associated pathways are not biologically relevant. For example, the type I IFN pathway is identified in all cell types, and can be biologically relevant in each of these cell types. Also the definition of ‘unique’ or ‘shared’ is somehow arbitrary, as it is depending on the resolution the cells have been classified. For example, you could classify cells as myeloid cells, monocytes or classical monocytes. Whether you would then classify a DE gene found in all myeloid cells as being unique, would depend on this classification.

2. Character overlaps in some supplementary figures, for example, Figure S5.

We have resolved the character overlap in Suppl Fig. S5 and Fig. 4.

Reviewer #3 (Remarks to the Author):

Oelen et al. present scRNA-seq analysis for PBMCs from a cohort of 120 individuals, and identify the presence of both cell-type-specific eQTLs and response eQTLs. This is an important work as so far there have been few single-cell population genetic studies aiming to find cell-type-specific eQTLs, and those that have been published profiled smaller populations. The analysis and results presented are well executed, and it was particularly interesting to see that response-specific eQTLs were often shared across the different pathogens, and decreased over time. I have a few comments which I hope can further improve the current manuscript.

A major goal of the study was to identify the cell types in which eQTLs are active. My main question is whether single-cell population genetic studies are required to determine the cell types implicated in eQTLs discovered from bulk-cell assays, or whether the cell types involved can be accurately inferred from other types of data. For example, for eQTL genes, you could find the list of cell types expressing that gene using single-cell RNA-seq data from a single individual. You could also look at scATAC-seq data from a single individual and examine which cell types had open ATAC peaks near or overlapping the variant position. In generating the single-cell population genetic dataset here, the authors are uniquely able to address this question, which I feel is valuable to the field: should we be pursuing single-cell population genetic experiments, or is it more effective to do targeted single-cell experiments on a small number of individuals and "deconvolve" signals found using bulk assays on much larger populations.

Deconvolution of bulk RNA-seq data has been used in the past to predict the cell type in which an eQTL effect takes place [Aran et al. 2017 Genome Biol; Newman et al. 2015 Nat Methods; Aguirre-Gamboa et al. 2020 - BMC informatics; de Klein et al. 2021 - biorxiv]. The accuracy of these deconvolution approaches is depending on how well they can predict the composition of a tissue. For whole blood the correlation between measured and predicted cell type proportions reaches up to $R = 0.73$ in the best predictable cell type [Aguirre-Gamboa et al. 2020 - BMC informatics], while for brain tissue it only reaches up to $R = 0.32$ [de Klein et al. 2021 - biorxiv]. On top of this, the predictability may be negatively affected when deconvolution is applied to conditions that were not seen in the training model, such as pathogen stimulations or disease conditions. Moreover, in the case that particular cell type proportions are highly correlating with each other (e.g. granulocytes and B cells had a correlation of 0.75 in the model used in Aguirre-Gamboa et al. [Aguirre-Gamboa et al. 2020 - BMC informatics]), false negative results can be the consequence in these cell types, especially for relatively weak eQTL effects.

It is important to realize that even though a particular gene is expressed in cell types, this does not always result in the same eQTL effect taking place. ATAC-seq data of the relevant cell type may help distinguish in which cell type the eQTL effect takes place, but the LD among SNPs could make it difficult to pinpoint the causal SNP, and as such it may be unclear which chromatin position to take into account when taking the ATAC-seq information into account.

And even if all of the above would be solved in the future, it is important to realize that deconvolution of bulk RNA-seq signal could be an alternative approach for identifying cell type-specific eQTLs, but not co-expression QTL analysis. Co-expression QTLs are calculated by making use of the many single cell observations that are available per individual, allowing the calculation of gene-gene correlations per individual. This is something that is not possible with a single 'bulk' measurement per individual. In contrast, interaction QTL analysis with a second gene as an interaction term can identify similar effects, although with much lower power and with the risk of identifying false co-expression relationships (as a result of Simpson's paradox) (van der Wijst et al. 2018; Simpson EH, 1951). As highlighted in our manuscript, co-expression QTLs are uniquely suitable to unbiasedly uncover the context in which gene regulation takes place, and therefore, together with the information above, it is for this purpose alone that it is valuable to generate population-level single-cell data.

We now discuss the above in the Results section (p16, line 326-332): “We have previously shown that genetics can influence the co-expression relationship between genes and that scRNA-seq data is uniquely suitable to do so by taking the individual cells per cell type per donor as observations over which the individual-specific co-expression is calculated.³² In contrast, bulk RNA-seq data usually contains a single measurement per donor, and therefore, co-expression in bulk data cannot be calculated at the individual level. As a consequence, the co-expression between two genes as calculated from bulk RNA-seq data may be different from the true individual-specific co-expression relationship as extracted from scRNA-seq data (due to Simpson’s paradox[Simpson EH, 1951, Van der Wijst 2018, Genome Medicine]).”

The analysis presented here has focused on six major PBMC cell types, but there are much more than this present in PBMCs (naive, memory CD4/CD8 T cells, pro/pre B cells, CD14/CD16 monocytes, etc.). Given the large sample size (across all donors), I am unsure why the authors have restricted their analysis to such broad cell type classifications. The major advantage of a single-cell assay is the ability to identify all of these different cell types in the dataset.

The reviewer is correct that the PBMCs could be classified with higher resolution than at the level that it was originally presented in the manuscript. One of the main reasons for us to focus our initial analyses on the six major cell types is that for eQTL analysis the number of cells per donor per (higher resolution sub)cell type determines the statistical power to detect effects. You can imagine that the calculated average expression per donor per cell type is more noisy when you have only 10 instead of 100 cells. As a consequence, it is also more difficult to determine whether an effect is subcell type-specific, as the subcell type of only 10 cells will likely have reduced power as opposed to the subcell type with 100 cells. Moreover, most gene expression and gene regulation effects are more likely shared between subcell types than different major cell types, so we expect most differences to be already detected at the major cell type level. Nevertheless, we agree that this higher resolution analysis could potentially provide additional insights that previously could not be studied. Therefore, we have now conducted additional DE (table S5), pathway (table S6) and eQTL (table S7) analyses in the following subcell types: 1. classical and non-classical monocytes, 2. Myeloid and plasmacytoid DCs, 3. NKdim and NKbright, 4. naive and memory CD4T cells, 5. naive and memory CD8T cells. In addition to the subcell type output that is now provided in table S5-S7, we now highlight the pathways that are most differentially activated across the two subcell types within each major cell type (figure S3) and present Venn diagrams showing the overlap in eGenes as identified in the two subcell types and their corresponding major cell type (figure S4c).

We now discuss the subcell type specific analysis in the Results section (p9, line 167-172): “A total of 5,516 unique DE genes were identified over all conditions and major cell types, and an additional 1,621 DE genes were identified in the subcell types (**Table S5**). This indicates that most DE genes can already be identified at the major cell type level. However, since the statistical power to detect such DE effects is correlated with the number of cells within a subcell type³², likely some of the subcell type specificity remains undetectable. Of the 5,516 DE genes within the major cell types, 31.1% were cell type-specific and 15.1% were shared across all major cell types (**Fig. 2b**). “

(p11, line 212-217): “For the subcell types, we were mainly interested in those pathways that were differentially activated upon pathogen stimulation between the two subcell types of each major cell type (**Fig. S3**). For this, we visualized the top 10 most enriched pathways with the largest difference in significance between both subcell types. This revealed that most pathways were enriched in both subtypes, but that the relative activation could differ. For example, several pathways associated with interferon signaling were more significantly enriched in the ncMono as opposed to the cMono (**Fig. S3**). “

(p12-13, line 257-261): “When overlapping the identified eGenes in the major cell types with each of their two subcell types (over all stimulation-timepoint combinations combined), we observed that the majority of eGenes identified in the subcell types were already detected in the corresponding major cell type (**Fig. S4c**). Nevertheless, 4.6% (for the NKdim) up to 24.5% (for the pDCs) additional eGenes were uniquely identified in such a subcell type.”

Minor comments:

Figure 2e is unclear (especially the black and white heat map at the top). Perhaps splitting genes by group and constructing separate heat maps would be more clear.

We have remade Fig. 2E now splitting genes by pathway and constructing separate heatmaps. This now provides a clearer picture on the uniformity or specificity of the pathways.

Figure 4b is unclear. I think the top plots are showing expression of CLEC12A vs PML, with each point being a cell, but I'm unsure why they're shown as lines originating at zero and why the data appear to be discrete on the x-axis?

We have remade Fig. 4B to ease the interpretation. First, we cleaned up unnecessary repetitions in the figure (‘rs12230244 on CLEC12A and PML’). Second, we have added subtitles to the top (‘individual-specific regression lines’) and bottom (‘Spearman correlation per individual’) plots stating what is shown.

In the top plots the expression of CLEC12A vs PML is shown, with each point being a cell. The lines show individual-specific regression lines through all these cells. As some cells do not show expression of CLEC12A and/or PML, some cells will be located at zero expression. The data on the x-axis shows SCT-normalized expression values (which are discrete). To clarify this, we now specify in the legend that normalized expression is shown (instead of referring to it as ‘expression’).

Several details in the single-cell RNA-seq analysis can be explained better. For example (line 593), how were integration anchors found? Wherever possible, the name of the function used and the parameters need to be reported.

We have now provided additional details or refer to additional relevant literature whenever this information was still lacking. The exact steps taken for Seurat’s integration approach are described at line 600-871 on the Github page accompanying the paper: https://github.com/molgenis/1M-cells/blob/master/seurat_preprocess_samples/1M_preprocess.R. In short, for each version chemistry-pathogen stimulation combination, the timepoints and the unstimulated condition were scaled by the total UMI count of a cell, and subsequently log-transformed. Integration anchors between the datasets were obtained using the FindIntegrationAnchors function in Seurat, using the first 30 principal components per condition. Actual integration was performed using the IntegrateData method in Seurat (as described in Stuart et al. 2019: [10.1016/j.cell.2019.05.031](https://doi.org/10.1016/j.cell.2019.05.031)).

We now describe this in more detail in the Methods section (p30, line 642-644): “For this, we used the first 30 principal components (PCs) to identify integration anchors using Seurat’s FindIntegrationAnchors function. These anchors were then used for integration using Seurat’s IntegrateData function. (Stuart et al. 2019).”

How were cell types annotated? What marker genes were used and what was the process of assigning cells to cell types based on these marker genes?

The integrated dataset's first 30 principal components (PCs) were selected to identify clusters representing cell types. For this, we constructed a k-nearest neighbor graph based on the euclidean distance in PCA space, and refined the edge weights between any two cells based on the shared overlap in their local neighborhoods (Jaccard similarity). This step was conducted using the FindNeighbors function in Seurat (k = 20) [Stuart et al. 2019] . The cells were then clustered using the Louvain algorithm using the FindClusters function in Seurat (resolution = 0.8) [Stuart et al. 2019]. Clusters were then visualized in 2D space using a UMAP, constructed from the first 30 PCs. Cell types were then annotated to each of those clusters using marker gene expression (Table S4). The integration approach in Seurat can be found in https://github.com/molgenis/1M-cells/blob/master/seurat_preprocess_samples/1M_preprocess.R from line 600 to 871.

We have now provided additional details or refer to additional relevant literature whenever this information was still lacking in paragraph 'scRNA-seq alignment, preprocessing and QC' (p30, line 639-648): *"For annotating the cell types, we first log-normalized the count matrices for each of the seven timepoint-stimulation conditions and two chemistries separately using Seurat's LogNormalize function (scale.factor = 10,000).⁷⁹ The log-normalized count matrices of the unstimulated data were then integrated separately for each of the three pathogen stimulations. For this, we used the first 30 principal components (PCs) to identify integration anchors using Seurat's FindIntegrationAnchors function. These anchors were then used for integration using Seurat's IntegrateData function.⁷⁹ We performed principal component analysis (PCA) and selected the first 30 PCs to identify the cell clusters using k-nearest neighbor clustering and visualized this in UMAP space (using the default settings). Cell types were assigned to each cluster based on marker gene expression, resulting in a set of six major cell types and ten subcell types (Fig. S1B, Table S4)."*

Where is the GeneticRiskScoreCalculator program available?

The GeneticRiskScoreCalculator program is available through Github (<https://github.com/molgenis/systemsgenetics/tree/master/GeneticRiskScoreCalculator>). We have now updated the Github repository associated with our manuscript to link to the GeneticRiskScoreCalculator program. Moreover, we now refer to the above Github page directly in the Methods section of the paper.

REVIEWER COMMENTS

Reviewer #1 (Remarks to the Author):

I thank the authors for their thorough responses to my comments and clarifications around their methods and approaches. My primary concern in review of the initial submission was around low coverage cells and various technical aspects of the manuscript and data analysis. I feel the author's have addressed these concerns in rebuttal and with the additional text they have provided in the manuscript.

My only remaining concern/comment with the manuscript is around the composition of the Lifelines cohort generally and the individuals selected from within this cohort for the current study specifically. As the authors note, Lifelines is a cohort population study of adults within the Northern Netherlands. The authors subselected from this cohort for the current study. Metadata for these subselected individuals are provided in Supp. Table 1 of the current submission. However, these data are limited to sex and age range only. I understand the need to limit the information within the Supp. Table to prevent the individuals from being potentially identifiable. However, my questions relate to how representative this cohort is of the broader global population and therefore how directly transferrable the finding so this study are to a broader population. This doesn't detract from the validity of the study itself, or the more general finding that there appear to be genetic QTLs that influence the responsiveness of specific cell populations to different pathogens. However, it understand the broader relevance of the work, I think the author's need to provide additional detail in their methods and supplementaries about the cohort and the subselected population used in the current study.

Specifically including data, where available on:

1. General ethnicity of the participants
2. Socioeconomics - which can be a marker of childhood development and the impact this has on
3. Evidence of or against prior known infection with any of the three pathogens being studied.
4. Known immunopathies within the individuals, or at least known genetic markers of these in the samples (acknowledging these markers are limited and this may not be possible/practical due to ethical clearance and permissions to access the cohort) that might be relevant to the study, or known prior familial history of these disorders (I suspect this will not be possible to gather post hoc). In the absence of this, general data on, to the extent known, the prevalence of relevant immunonological disorders within the Netherlands, would at least provide some information.
5. Smoking history and other lung ailments (e.g., asthma/allergies). I note ~20% of the participants in the Lifelines cohort as described in Tigchelaar et al are smokers. In addition, ~20% have IBS. Have these attributes been explored as potential confounding factors? This is particularly relevant when considering respiratory infections.

Ideally, if the authors can access some of the above information, they can assess whether any of these factors correlate with the responses they see (or rule this out). In the absence of these data, I think the authors need to provide additional comment in the discussion around the potential limitations in the study in terms of applying their findings to other populations that may have a very different genetic composition.

Reviewer #2 (Remarks to the Author):

After carefully reading of the revision version of the manuscript. I think it has been improved greatly; the authors answered all my concerns. Still, their conclusions would be more convincing if experimental evidence could be provided.

Reviewer #3 (Remarks to the Author):

The authors have addressed my initial comments. I only have one minor comment on the revision: in the code linked by the authors, CCA is used for integration, but the methods text states that PCA used.

Response to reviewer comments

We would like to thank the reviewers for their highly constructive comments and suggestions, which have enabled us to improve our manuscript substantially. Below we will provide a response (plain text) to each reviewer comment (bold) with specific adjustments to the text being italicized.

Reviewer #1 (Remarks to the Author):

thank the authors for their thorough responses to my comments and clarifications around their methods and approaches. My primary concern in review of the initial submission was around low coverage cells and various technical aspects of the manuscript and data analysis. I feel the author's have addressed these concerns in rebuttal and with the additional text they have provided in the manuscript.

My only remaining concern/comment with the manuscript is around the composition of the Lifelines cohort generally and the individuals selected from within this cohort for the current study specifically. As the authors note, Lifelines is a cohort population study of adults within the Northern Netherlands. The authors subselected from this cohort for the current study. Metadata for these subselected individuals are provided in Supp. Table 1 of the current submission. However, these data are limited to sex and age range only. I understand the need to limit the information within the Supp. Table to prevent the individuals from being potentially identifiable. However, my questions relate to how representative this cohort is of the broader global population and therefore how directly transferrable the finding so this study are to a broader population. This doesn't detract from the validity of the study itself, or the more general finding that there appear to be genetic QTLs that influence the responsiveness of specific cell populations to different pathogens. However, it understand the broader relevance of the work, I think the author's need to provide additional detail in their methods and supplementaries about the cohort and the subselected population used in the current study.

Specifically including data, where available on:

1. General ethnicity of the participants

The vast majority of the Lifelines cohort is born in the Netherlands (97%) and has a Caucasian ethnicity (98%). Specifically, the 120 individuals that were included in this study were all Caucasian.

We now mention this in the methods section (p28, line 597-598): "*Whole blood from 120 Caucasian individuals of the northern Netherlands population cohort Lifelines Deep⁷⁸ was drawn into EDTA-vacutainers (BD).*"

2. Socioeconomics - which can be a marker of childhood development and the impact this has on

We have updated Table S1 with additional donor metadata, now also including an individual's Neighborhood socio-economic status score (nSES) and highest educational degree.

3. Evidence of or against prior known infection with any of the three pathogens being studied.

Regretfully, this data is not available in our cohort.

4. Known immunopathies within the individuals, or at least known genetic markers of these in the samples (acknowledging these markers are limited and this may not be possible/practical due to ethical clearance and permissions to access the cohort) that might be relevant to the study, or

known prior familial history of these disorders (I suspect this will not be possible to gather post hoc). In the absence of this, general data on, to the extent known, the prevalence of relevant immunological disorders within the Netherlands, would at least provide some information.

Regretfully, this data is not available in our cohort. We have, however, now supplied an additional tab to Table S1 ('DutchPopulation') that contains the prevalence of relevant immune diseases in the Dutch population.

5. Smoking history and other lung ailments (e.g., asthma/allergies). I note ~20% of the participants in the Lifelines cohort as described in Tigchelaar et al are smokers. In addition, ~20% have IBS. Have these attributes been explored as potential confounding factors? This is particularly relevant when considering respiratory infections.

We have updated Table S1 with additional donor metadata, now also including smoking history and lung ailments. Moreover, we describe the prevalence of several immune-mediated diseases in the Dutch population in an additional tab ('DutchPopulation').

Reviewer #2 (Remarks to the Author):

After carefully reading of the revision version of the manuscript. I think it has been improved greatly; the authors answered all my concerns. Still, their conclusions would be more convincing if experimental evidence could be provided.

Reviewer #3 (Remarks to the Author):

The authors have addressed my initial comments. I only have one minor comment on the revision: in the code linked by the authors, CCA is used for integration, but the methods text states that PCA used.

The FindAnchors function by default uses a Canonical Correlation Analysis to get the anchors required to subsequently integrate the data using Seurat's IntegrateData. For the integration, we used the default of the first 30 components. After integration, PCA is used to obtain the principal components of the combined set, that are used for the KNN clustering and UMAP dimensionality reduction.

We have updated the text in the methods section, to better clarify what was used in each step (p30, line 642-644): *"For this, we used the first 30 dimensions from a Canonical Correlation Analysis to identify integration anchors in Seurat's FindIntegrationAnchors function."*

REVIEWERS' COMMENTS

Reviewer #1 (Remarks to the Author):

Thank you to the authors for their considered responses to my comments. I think these changes are good additions.

I am sorry to continue to make further suggestions as I do not wish to unfairly delay decision on the manuscript. However, I do think the points the authors raise on the ethnic composition of the cohort (all Caucasian) and the unavailability of potentially relevant prior exposures to the pathogens being studied or possible immunopathies that could confound the results should be addressed in some form.

In my opinion, it would be sufficient to make clear in the abstract and discussion, that this cohort is solely caucasians and the results may not be reflective of behaviour in other cohorts. It is unfortunately the authors aren't able to access the medical history data noted above, but I understand that there is nothing that can be done about that. As an alternative, I suggest the authors provide, if available, reference to data on the prevalence of *C. albicans*, *P. aeruginosa* and *M. tuberculosis* infection in the Netherlands as a whole or (although I imagine the data is not available) specific to the northern Netherlands region. Similarly, unfortunately I am not an immunologist, so I can't suggest which, in any, might be relevant, but if the authors know of any specific immunopathies that might impact on their results and have national/regional prevalence data for this in the Netherlands, I think that too should be referenced and noted in the discussion. If these data are not available in any form, then I'd ask the authors to note this in the discussion. If there are no known immunopathies that would give this result, then that's fine.

I have no further comments on the manuscript and congratulate the authors on their very interesting study.

Reviewer #3 (Remarks to the Author):

The authors have addressed all my comments

Response to reviewer comments

We would like to thank the reviewers for their remaining input. Below we will provide a response (plain text) to each reviewer comment (bold) with specific adjustments to the text being italicized.

Reviewer #1 (Remarks to the Author):

Thank you to the authors for their considered responses to my comments. I think these changes are good additions.

I am sorry to continue to make further suggestions as I do not wish to unfairly delay decision on the manuscript. However, I do think the points the authors raise on the ethnic composition of the cohort (all Caucasian) and the unavailability of potentially relevant prior exposures to the pathogens being studied or possible immunopathies that could confound the results should be addressed in some form.

In my opinion, it would be sufficient to make clear in the abstract and discussion, that this cohort is solely caucasians and the results may not be reflective of behaviour in other cohorts.

We now discuss this point in the Discussion section (p27, line 594-597): *“Importantly, this study was conducted in European individuals with a white background. Although we do not expect general conclusions to be different in other populations, it may be that the upstream regulators or downstream consequences of some of the specific genetic variants act differently across populations.”*

It is unfortunately the authors aren't able to access the medical history data noted above, but I understand that there is nothing that can be done about that. As an alternative, I suggest the authors provide, if available, reference to data on the prevalence of *C. albicans*, *P. aeruginosa* and *M. tuberculosis* infection in the Netherlands as a whole or (although I imagine the data is not available) specific to the northern Netherlands region. Similarly, unfortunately I am not an immunologist, so I can't suggest which, in any, might be relevant, but if the authors know of any specific immunopathies that might impact on their results and have national/regional prevalence data for this in the Netherlands, I think that too should be referenced and noted in the discussion. If these data are not available in any form, then I'd ask the authors to note this in the discussion. If there are no known immunopathies that would give this result, then that's fine.

General prevalence of these type of infections is difficult to obtain, as they do not necessarily lead to (severe) symptoms in non-immunocompromised individuals. We have included an additional sentence on this point in the Discussion section (p27, line 597-599): *“Moreover, as the infection history with the three pathogens under study is unknown for the individuals included in our study, there is a small chance that this may have introduced additional noise or confounding in our analyses.”*

I have no further comments on the manuscript and congratulate the authors on their very interesting study.

Reviewer #3 (Remarks to the Author):

The authors have addressed all my comments